# Compositional Causal Reasoning Evaluation in Language Models

**Jacqueline R. M. A. Maasch** [1]   **Alihan Hüyük** [2]   **Xinnuo Xu** [3]   **Aditya V. Nori** [3]   **Javier Gonzalez** [3]

## Abstract

Causal reasoning and compositional reasoning are two core aspirations in AI. Measuring the extent of these behaviors requires principled evaluation methods. We explore a unified perspective that considers both behaviors simultaneously, termed *compositional causal reasoning* (CCR): the ability to infer how causal measures compose and, equivalently, how causal quantities propagate through graphs. We instantiate a framework for the systematic evaluation of CCR for the average treatment effect and the probability of necessity and sufficiency. As proof of concept, we demonstrate CCR evaluation for language models in the Llama, Phi, and GPT families. On a math word problem, our framework revealed a range of taxonomically distinct error patterns. CCR errors increased with the complexity of causal paths for all models except o1.

## 1. Introduction

*Causal reasoning* is a defining outcome of human evolution (Goddu & Gopnik, 2024). Humans flexibly reason about cause and effect in factual realities that can be observed and intervened on, as well as imagined counterfactual worlds. A causal lens enables humans and machines alike to learn generalizable lessons about the mechanics of the universe (Schölkopf et al., 2021). Thus, human-like AI might require reasoning at all three levels of Pearl's Causal Hierarchy: associational, interventional, and counterfactual (Pearl, 2000).

Human-like AI might also require *compositional reasoning* (Lake et al., 2017): the capacity to recognize and synthesize novel combinations of previously observed concepts (Xu et al., 2022). Compositionality is ubiquitous in the physical world, symbolic systems, and human cognition

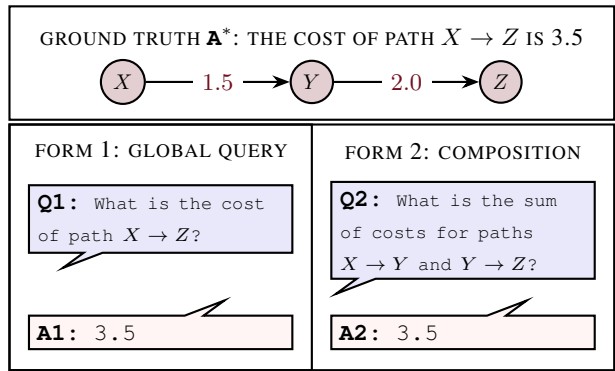

Figure 1. Compositionally consistent responses to two formulations of a simple (non-causal) query. Reasoning is internally consistent if A1==A2, and externally valid if A1==$A^*$ and A2==$A^*$.

(Frankland & Greene, 2020), underlying both visual perception (Schwartenbeck et al., 2023) and language (Lake & Baroni, 2023).[1] It is both a means of generalization and of coping with complexity: problems can be reformulated as simpler subproblems connected by compositional rules.

The present work explores causal and compositional reasoning in tandem. We center our focus on reasoning evaluation in language models (LMs), given increasing interest in LM reasoning emergence (Huang & Chang, 2023; Qiao et al., 2023; Mialon et al., 2023). Following from traditions in causal inference and graphical modeling, we define *compositional causal reasoning* (CCR) as

> *the ability to infer compositions and decompositions of causal measures in factual and counterfactual worlds.*

By extension, this requires reasoning over the propagation of causal quantities through graphs. To facilitate CCR evaluation, we introduce a framework for the exhaustive assessment of *compositional consistency*: correct inference that equivalent compositions are indeed equal. We measure compositional consistency with respect to ground truth (*external validity*) and concordance among the LM's responses (*internal consistency*) (Fig. 1). We empirically demonstrate instantiations of our framework for two causal measures: the average treatment effect (ATE) and the probability of necessity and sufficiency (PNS; Pearl 1999), which coincides with the ATE in certain data generating processes.

[1]Cornell Tech [2]Harvard University [3]Microsoft Research Cambridge. Correspondence to: Jacqueline Maasch <maasch@cs.cornell.edu>. Work done while authors J. Maasch and A. Hüyük were interns at Microsoft Research Cambridge.

*Proceedings of the 42nd International Conference on Machine Learning*, Vancouver, Canada. PMLR 267, 2025. Copyright 2025 by the author(s).

---

[1]See Fig. A.1 for examples.

### 1.1. Contributions

§3 **A compositional view of causal reasoning in LMs.** We formally express CCR as the ability of an LM to infer causal measure compositions (*inductive* reasoning) and decompositions (*deductive* reasoning).

§4 **Taxonomy of reasoners.** We propose measures of external validity and internal consistency for compositional consistency evaluation. To facilitate error analyses, we introduce a taxonomy of reasoning patterns: *valid-consistent* (VC), *valid-inconsistent* (VI), *invalid-consistent* (IC), and *invalid-inconsistent* (II).

§5 **Evaluation framework and task generator.** We introduce a procedure for evaluating inductive CCR for the ATE and PNS in causal graphs with cutpoints (Alg. 1). To facilitate future work, we provide open-source code for randomly generating qualitative and quantitative CCR tasks of scalable graphical complexity.[2]

§6 **Empirical demonstration.** We deploy Alg. 1 to evaluate CCR in seven LM architectures, with and without chain-of-thought (CoT) prompting. Even on a simple CCR problem, our framework revealed taxonomically distinct patterns of inconsistent and invalid reasoning, ranging from II to VC.

### 1.2. Related Works

**LM Reasoning** This work contributes to the theoretical and empirical study of compositional (Hudson & Manning, 2019; Hupkes et al., 2020; Xu et al., 2022; Ito et al., 2022; Hsieh et al., 2023), causal (Wang et al., 2023c; Du et al., 2022), mathematical (Saxton et al., 2019; Lewkowycz et al., 2022; Stolfo et al., 2023; Lu et al., 2023b), inductive (Qiu et al., 2024), and graphical reasoning (Wang et al., 2023a; He et al., 2024) in AI. Compositional reasoning has been examined in various neural architectures, including transformers (Li et al., 2023; Lu et al., 2023a; Dziri et al., 2023) and diffusion models (Du et al., 2023; Okawa et al., 2023; Su et al., 2024). In this work, we focus on transformer-based autoregressive LMs. This work introduces a new compositional viewpoint on *consistency*, an ongoing concern in LM research (Wang et al., 2023b; Li et al., 2024).

Despite some optimistic results (Kıcıman et al., 2023), several recent works hypothesize that current LMs are not capable of true logical reasoning (Mirzadeh et al., 2025) and are merely *causal parrots* (Zečević et al., 2023). Counterfactual reasoning in LMs can be brittle (González & Nori, 2024), and performance on formal causal reasoning tasks can decline monotonically with task difficulty (Jin et al., 2023). Similarly, multiple works find that models struggle with compositional reasoning in vision and language, especially for complex compositions (Agrawal et al., 2017; Ma et al., 2023; Ray et al., 2023; Press et al., 2023). Efforts

[2]Project page: https://jmaasch.github.io/ccr/

to elicit causal, compositional, and mathematical reasoning via fine-tuning (Hüyük et al., 2025) or CoT prompting (Wei et al., 2022; Jin et al., 2023; Press et al., 2023) have shown promising yet limited success. A survey of causal reasoning benchmarks found that most suffer from critical design flaws (e.g., data contamination and inappropriate evaluation metrics), highlighting the need for improved evaluation frameworks (Yang et al., 2024). To our knowledge, prior frameworks do not explicitly and systematically incorporate causal compositionality. See Appendix A for additional related works. For an in-depth comparison to prior causal reasoning evaluation frameworks, see Table A.1.

**Compositionality & Causality** While explicitly combining compositional and causal reasoning evaluation in LMs is new, the intersection of compositionality and causality already enjoys a rich mathematical framework: the formalisms and methods of graphical modeling and causal inference. A central concern of probabilistic and causal graphical modeling is factorization: the expression of complex multivariate distributions as products (or *compositions*) of local distributions, enabling efficient learning and inference (Koller & Friedman, 2009). Graphs provide expressive representations for joint distributions, their factors, and the propagation of quantities through systems (Pearl, 1982; Shafer & Shenoy, 1990; Kschischang et al., 2001). In causal inference, the decomposition of causal effects (Avin et al., 2005; Pearl, 2014; VanderWeele, 2014; Singal & Michailidis, 2024) plays a central role in mediation analysis (VanderWeele, 2016), fairness analysis (Plečko & Bareinboim, 2024), and covariate adjustment in the presence of latent variables (Pearl, 1995; Jeong et al., 2022). These traditions offer a convenient mathematical language for evaluating compositional and causal reasoning simultaneously.

## 2. Preliminaries

**Notation** Uppercase denotes univariate random variables (e.g., $Y$) and bold uppercase denotes sets or multivariate random variables (e.g., $\mathbf{V}$), with realizations in lowercase (e.g., scalars $x$; vector values $\mathbf{x}$). Models and graphs are denoted by calligraphic script (e.g., causal graph $\mathcal{G}$). For $X \in \mathcal{G}$, we denote the parents of $X$ by $\mathbf{pa}_X$.

### 2.1. Causal Models

Structural causal models (SCMs; Pearl 2009) provide a convenient coupling for our framework: (1) a rich mathematical language for expressing the compositionality of causal measures and (2) an intuitive visual language in the form of directed acyclic graphs (DAGs).

**Definition 2.1** (Structural causal model (SCM), Pearl 2001)**.** An SCM is a tuple $\mathcal{M} \coloneqq \langle \mathbf{V}, \mathbf{U}, \mathcal{F}, p(\mathbf{u}) \rangle$, where $\mathbf{U} = \{U_i\}_{i=1}^n$ is a set of exogenous variables determined by factors outside $\mathcal{M}$, $\mathbf{V} = \{V_i\}_{i=1}^n$ is a set of observed en-

dogenous variables determined by variables in $\mathbf{U} \cup \mathbf{V}$, $\mathcal{F} = \{f_i\}_{i=1}^{n}$ is a set of structural functions such that $v_i = f_i(\mathbf{pa}_{v_i}, u_i)$, and $p(\mathbf{u})$ is the distribution over $\mathbf{U}$.

We restrict our attention to *positive-Markovian* SCMs whose graphical representations are DAGs.

**Definition 2.2** (Positive-Markovian SCM, Pearl 1999). An SCM is Markovian if its graphical representation is acyclic and exogenous variables $U_i$ are mutually independent. An SCM is positive-Markovian if it is Markovian and $\mathbb{P}(v) > 0$ for every realization $V \in \mathbf{V} = v$.

Let $X, Y \in \mathcal{M}$ be binary random variables. Let $pa_X$ denote the realization of a parent of $X$. Under Def. 2.2, the causal effect of $X$ on $Y$ is identifiable as (Pearl, 1999)

$$\mathbb{P}(Y = y \mid do(X = x)) = \sum_{pa_X} \mathbb{P}(y \mid x, pa_X)\mathbb{P}(pa_X). \quad (1)$$

### 2.2. Causal Measures

**Average Treatment Effect**  The most widely studied causal estimand (Imbens, 2004), the ATE measures the effect of receiving treatment versus no treatment on the mean outcome over the population.

**Definition 2.3** (Average treatment effect (ATE)). Let $X$ denote a binary treatment variable and $Y$ an outcome. We express the ATE as the following difference of expectations:

$$\text{ATE} := \mathbb{E}[Y \mid do(X = 1)] - \mathbb{E}[Y \mid do(X = 0)]. \quad (2)$$

**Probability of Necessity and Sufficiency**  In propositional logic, we say that $X$ is *necessary* for $Y$ when $Y \implies X$, $X$ is *sufficient* for $Y$ when $X \implies Y$, and $X$ is *necessary and sufficient* for $Y$ when $X \iff Y$. Pearl (1999) introduced a probabilistic framework for reasoning over necessity and sufficiency with the *probabilities of causation* (PrC): the probabilities of necessity (PN), sufficiency (PS), and necessity and sufficiency (PNS). In the present work, we focus on the PNS.

Let $X$ and $Y$ denote binary random variables, where $X$ is a cause of $Y$. Let $x$ and $y$ denote the *propositions* or *events* that $X = \text{TRUE}$ and $Y = \text{TRUE}$, respectively, while $x'$ and $y'$ denote that $X = \text{FALSE}$ and $Y = \text{FALSE}$.

**Definition 2.4** (Probability of necessity and sufficiency (PNS), Pearl 1999). The probability that $x$ is necessary and sufficient to produce $y$ is given as

$$\text{PNS} := \mathbb{P}(y_x, y'_{x'}) = \mathbb{P}(x, y)\text{PN} + \mathbb{P}(x', y')\text{PS}. \quad (3)$$

The PNS is point identifiable from causal effects when $Y$ is monotonic in $X$: changing $X$ from FALSE to TRUE does not induce $Y$ to change from TRUE to FALSE.

**Definition 2.5** (Identification of the PNS, Tian & Pearl 2000). Given a positive-Markovian SCM (Def. 2.2) for which $Y$ is monotonic in $X$, the PNS is given as

$$\mathbb{P}(y_x) - \mathbb{P}(y_{x'}) = \mathbb{P}(y \mid do(x)) - \mathbb{P}(y \mid do(x')) \quad (4)$$

where effects $\mathbb{P}(y_x)$ and $\mathbb{P}(y_{x'})$ are identifiable by Eq. 1. Note that Eq. 4 is equivalent to the ATE (Proposition B.4). We will leverage this fact for CCR evaluation in Section 5.

## 3. Compositional Causal Reasoning

Compositionality has been variously defined in linguistics (Haugeland, 1979; Wittgenstein, 2009), category theory (Fong & Spivak, 2019), quantum theory (Coecke, 2023), and other domains. For example, while a Schrödinger compositional theory dictates that compositions are "greater than the sum of their parts" (Coecke, 2023), such emergent effects are not relevant in our setting. We highlight this variation to clarify that measuring compositional reasoning in AI is contingent on a chosen definition of compositionality. We select a function-based viewpoint that draws from probabilistic graphical modeling and causal inference.[3]

**Definition 3.1.** *Compositionality* exists when a measure $f$ can be expressed as a function of measures $\{g_i\}_{i=1}^{n \geq 2}$.

This definition is intentionally lax to capture a wide range of mathematical behavior. It allows for any function to serve as the compositional rule (e.g., addition, multiplication, etc.). As in probabilistic graphical modeling, we refer to compositions $f$ as *global* measures and to constituent $g_i$ as *local* measures. These are terms of relative scale: a local measure in one setting may also admit decompositions at finer granularity. Thus, compositionality implies a scale of interest (as exemplified by the physics example in Fig. A.1). Def. 3.1 arises frequently in causal inference, e.g.:

*Example* 3.2 (Decomposition of total causal effects in linear SCMs, Pearl 2001). Let TE be the total effect, NDE the natural direct effect, and NIE the natural indirect effect. When causal functions are linear,

$$\underbrace{\text{TE}}_{global} = \underbrace{\text{NDE}}_{local} + \underbrace{\text{NIE}}_{local}. \quad (5)$$

Following from Definition 3.1, we define a compositional interpretation of causal reasoning in AI.

**Definition 3.3** (Compositional causal reasoning (CCR)). The ability to correctly infer (1) how local causal measures *compose* into global causal measures and (2) how global causal measures *decompose* into local causal measures, in both factual and counterfactual worlds.

---

[3]Our definition is similar to *decomposability* as given by Def. 3.5 in Plečko & Bareinboim (2024).

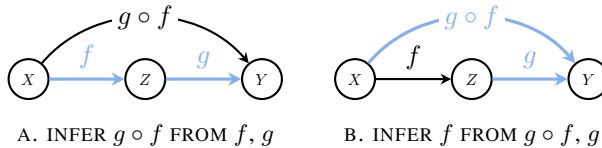

Figure 2. (**A**) Inductive and (**B**) deductive CCR.

Definition 3.3 encompasses reasoning over both compositions and decompositions, which we can disaggregate into *inductive* and *deductive* reasoning (Fig. 2).

**Definition 3.4** (Inductive CCR). The ability to reason about the *composition* of causal measures. With respect to Def. 3.1, this corresponds to inferring composition $f$ given knowledge of local measures $\{g_i\}_{i=1}^{n \geq 2}$.

**Definition 3.5** (Deductive CCR). The ability to reason about the *decomposition* of causal measures. With respect to Def. 3.1, this corresponds to inferring local measure $g_j$ given knowledge of composition $f$ and measures $\{g_i\}_{i=1}^{n \geq 2} \setminus g_j$.

With respect to Example 3.2, inductive CCR entails inferring TE from NDE and NIE, while deductive CCR entails inferring NDE from TE and NIE (etc.).

## 4. Compositional Consistency Evaluation

### 4.1. Obtaining Causal Estimates

Let $\mathcal{A}$ denote a model (e.g., an LM), $\Phi$ the set of all causal measures, and $\varphi \in \Phi$ a measure of interest that we seek to evaluate for variables $\mathbf{V}$ in SCM $\mathcal{M}$ (e.g., the ATE). Let $\varphi_{\mathbf{x}}$ be a *causal query* about the value of $\varphi$ with respect to (w.r.t.) $\mathbf{X} \subseteq \mathbf{V}$. We denote the true value of $\varphi_{\mathbf{x}}$ by $\varphi_{\mathbf{x}}^*$.

Causal estimates $\widehat{\varphi}_{\mathbf{x}}$ can be obtained by various means, including (1) explicitly querying $\mathcal{A}$ for the value of $\varphi_{\mathbf{x}}$, requiring $\mathcal{A}$ to directly perform formal causal inference (as in Jin et al. 2023); or (2) implicitly querying $\mathcal{A}$ for the value of $\varphi_{\mathbf{x}}$, where responses are used to perform formal causal inference downstream of $\mathcal{A}$ (as in González & Nori 2024; Hüyük et al. 2025). In either case, each causal query is encoded in a *question template* (or series of templates)

$$\mathcal{Q}_{\varphi_{\mathbf{x}}} := (\varphi_{\mathbf{x}}, \mathcal{S}), \tag{6}$$

where $\mathcal{S}$ is some surface form that expresses accessory details (e.g., the background of a math word problem) (Stolfo et al., 2023). $\mathcal{Q}_{\varphi_{\mathbf{x}}}$ is expressed in a form comprehensible to $\mathcal{A}$ (e.g., text, image, etc.) and can either directly state $\varphi_{\mathbf{x}}$ (as in case 1) or logically imply it (as in case 2). Causal estimates are then obtained by

$$\widehat{\varphi}_{\mathbf{x}} := f(\mathcal{A}(\mathcal{Q}_{\varphi_{\mathbf{x}}})), \tag{7}$$

where $f$ is some transformation of the raw response from $\mathcal{A}$. In case (1), $f$ might be the identity function. One possible

formulation of case (2) defines $\mathcal{Q}_{\varphi_{\mathbf{x}}}$ as a series of questions, where $\widehat{\varphi}_{\mathbf{x}}$ is obtained by a transformation on the vector of responses. We demonstrate the latter formulation in Sec. 6.

### 4.2. Measuring Compositional Consistency

In this work, evaluation is performed w.r.t. a *task-metric-model triple* $\langle \mathcal{T}, \theta, \mathcal{A} \rangle$ for CCR task $\mathcal{T}$, some error metric $\theta$, and model $\mathcal{A}$ (Schaeffer et al., 2023).

**Definition 4.1** (CCR task). Let $\mathcal{M}$ denote the SCM representing the problem structure and $\varphi$ the estimand of interest. Let $\mathcal{Q} := \{\mathcal{Q}_{\varphi_{\mathbf{x}}}\}_{i=1}^n$, where causal queries $\{\varphi_{\mathbf{x}}\}_{i=1}^n$ correspond to the $n$ cause-effect pairs of interest in $\mathcal{M}$. We define the corresponding CCR task as the tuple

$$\mathcal{T} := \langle \varphi, \mathcal{M}, \mathcal{Q} \rangle. \tag{8}$$

Note that multiple $\varphi$ may be identifiable for $\mathcal{M}$, infinitely many $\mathcal{Q}$ can map to $\langle \varphi, \mathcal{M} \rangle$, and success of $\mathcal{A}$ on $\langle \varphi, \mathcal{M}, \mathcal{Q} \rangle$ does not imply success on $\langle \varphi, \mathcal{M}, \mathcal{Q}' \rangle$, $\langle \varphi, \mathcal{M}', \mathcal{Q}' \rangle$, $\langle \varphi', \mathcal{M}, \mathcal{Q}' \rangle$, etc. The appropriate choice of $\theta$ will be context-dependent (e.g., mean squared error, etc.).

We now explore a prime benefit of compositional perspectives on reasoning: the ease of introducing notions of *consistency*. The following concept of consistency can be applied to any form of reasoning, not only causal.

**Definition 4.2** (Compositional consistency). Reasoning is *compositionally consistent* when theoretically equivalent compositions are inferred to be equal.

Under the umbrella of compositional consistency, we can quantify CCR in many ways. For example, a model that succeeds at task $\mathcal{T}$ should not be prone to false negatives (i.e., failing to infer compositionality when it exists) nor false positives (i.e., hallucinating compositionality when it does not exist). In this work, we emphasize the *external validity* and *internal consistency* of CCR.

**Definition 4.3** (External validity). Reasoning is *externally valid* when estimates are equivalent to ground truth, up to some error $\delta$:

$$\theta(\varphi_{\mathbf{x}}^*, \widehat{\varphi}_{\mathbf{x}}) \leq \delta. \tag{9}$$

In Example 3.2, externally valid reasoning for cause-effect pair $\{X, Y\}$ entails that the following are below some threshold: $\theta(\text{TE}_{XY}^*, \widehat{\text{TE}}_{XY})$, $\theta(\text{NDE}_{XY}^*, \widehat{NDE}_{XY})$, $\theta(\text{TE}_{XY}^*, \widehat{\text{NDE}}_{XY} + \widehat{\text{NIE}}_{XY})$, etc.

**Definition 4.4** (Internal consistency). Reasoning is *internally consistent* when quantities that are theoretically equivalent are inferred to be equivalent, up to some error $\delta$:

$$\varphi_{\mathbf{x}}^* = \varphi_{\mathbf{x}'}^* \implies \theta(\widehat{\varphi}_{\mathbf{x}}, \widehat{\varphi}_{\mathbf{x}'}) \leq \delta. \tag{10}$$

|  | CONSISTENT | INCONSISTENT |
|---|---|---|
| **VALID** | **Q:** $(2 \mid 12) \wedge (3 \mid 12)$ ?  **A:** TRUE.
**Q:** $(6 \mid 12)$ ?  **A:** TRUE. | ✗ |
| **INVALID** | **Q:** $(2 \mid 12) \wedge (3 \mid 12)$ ?  **A:** FALSE.
**Q:** $(6 \mid 12)$ ?  **A:** FALSE. | **Q:** $(2 \mid 12) \wedge (3 \mid 12)$ ?  **A:** FALSE.
**Q:** $(6 \mid 12)$ ?  **A:** TRUE. |

DIVISIBILITY RULE:

If a number $n$ is divisible by 2 and 3, then it is divisible by 6. That is, $(2 \mid n) \wedge (3 \mid n) \Longrightarrow (6 \mid n)$.

*Figure 3.* VC, VI, IC, and II reasoning for a logic problem applying divisibility rules. For such a problem, we see that VI reasoning is impossible. However, all four types of reasoners can arise in the probabilistic setting in which we evaluate LMs.

Here, estimates are compared to each other rather than ground truth. In Example 3.2, internally consistent reasoning requires that $\theta(\widehat{\text{TE}}_{XY}, \widehat{NDE}_{XY} + \widehat{NIE}_{XY}) \leq \delta$.

**Definition 4.5** (Completeness). Given threshold $\delta$, reasoning is *complete* w.r.t. $\langle \varphi, \mathcal{M}, \mathcal{Q} \rangle$ when the following holds:

$$\forall \, \varphi_{\mathbf{x}} \, : \, \theta(\varphi_{\mathbf{x}}^{*}, \widehat{\varphi}_{\mathbf{x}}) \leq \delta \ \text{ and} \tag{11}$$

$$\forall \, \varphi_{\mathbf{x}}, \varphi_{\mathbf{x}'} \, : \, \varphi_{\mathbf{x}}^{*} = \varphi_{\mathbf{x}'}^{*} \Longrightarrow \theta(\widehat{\varphi}_{\mathbf{x}}, \widehat{\varphi}_{\mathbf{x}'}) \leq \delta. \tag{12}$$

**Taxonomy of Reasoners**  Following from Defs. 4.3 and 4.4, we delineate four categories of reasoners for task $\mathcal{T}$:

1. *Valid-consistent* (VC).  3. *Invalid-consistent* (IC).
2. *Valid-inconsistent* (VI).  4. *Invalid-inconsistent* (II).

Following from this taxonomy, there are three distinct profiles of incomplete CCR (IC, VI, II) but only one profile that can achieve completeness (VC). To illustrate, consider the logic problem given by Fig. 3:

*Example* 4.6. Consider a logic problem applying the divisibility rule $[(2 \mid n) \wedge (3 \mid n) \Longrightarrow (6 \mid n)]$ where $n = 12$. A reasoner can respond TRUE or FALSE to each query.

1. **VC**: Each individual query is answered correctly (externally valid) and the logical implication TRUE $\Longrightarrow$ TRUE is correct (internally consistent).
2. **IC**: Individual queries are answered incorrectly (externally invalid), but the logical implication FALSE $\Longrightarrow$ FALSE is correct (internally consistent).
3. **II**: One individual response is wrong (externally invalid) and the logical implication FALSE $\Longrightarrow$ TRUE is incorrect (internally inconsistent).

We see that only VC, IC, and II arise in this logic setting. However, LM evaluation is a probabilistic setting where errors are thresholded. Thus, all four types of reasoners can arise in LM evaluation, as illustrated in Section 6.

**Implementation**  This conceptual introduction has left implementation details intentionally vague. In theory, the framework proposed here can be implemented for any compositional formulae in causal inference (e.g., Example 3.2).

Section 5 suggests one possible procedure for assessing compositional consistency in causal reasoning (Alg. 1), based on compositional properties of the ATE and PNS in graphs with cutpoints. Section 6 provides an empirical illustration in LMs, demonstrating proof of viability.

## 5. Inductive CCR Evaluation for the PrC

### 5.1. PNS Composition Across Graph Components

The PNS boasts several convenient properties for reasoning evaluation: (1) variables of interest are binary and probabilities are bounded by 0 and 1; (2) translating PrC queries to text prompts designed to elicit logical, mathematical, probabilistic, and/or causal reasoning is relatively straightforward (González & Nori, 2024; Hüyük et al., 2025); and (3) the PNS and ATE coincide under certain conditions (Proposition B.4), and thus share convenient compositional properties.

**Assumptions: DAGs with Cutpoints**  In this work, we derive compositional forms for the PrC in graphs with cutpoints. A *cutpoint*, *cut vertex*, or *articulation point* is any node contained in multiple *biconnected components* (BCCs): maximal biconnected subgraphs induced by a partition of edges, such that two edges are in the same partition if and only if they share a common simple cycle (Westbrook & Tarjan, 1992). Thus, removing a cutpoint disconnects the graph. For ease of exposition, we demonstrate our framework on SCMs whose causal DAGs $\mathcal{G}_{XY}$ contain the following: (**A1**) only one root node $X$ (i.e., the cause of interest), (**A2**) only one leaf node $Y$ (i.e., the effect of interest), (**A3**) at least one cutpoint, and (**A4**) no unobserved confounders for cause-effect pairs of interest. Thus, we assume models with a single "source" node whose causal influence follows multiple indirect pathways to a single "sink" node. See Figs. 4, B.1, and D.1 for DAGs that satisfy A1–A4. Note that alternative formulations of this framework could relax these assumptions (e.g., Appendix C explores violations of A3).

**PNS Compositionality**  Though the PN and PS have proven useful for AI reasoning evaluation (González & Nori, 2024; Hüyük et al., 2025), we prove in Appendix B.2 that their composition across BCCs is complex. However, both

**Algorithm 1** *Inductive CCR eval. in graphs with cutpoints*

**Input:** CCT $\mathcal{C}_{XY}$; estimates $\{\widehat{\varphi}\cdot\}$, true values $\{\varphi^*\cdot\}$ for $\langle\varphi, \mathcal{M}, \mathcal{Q}\rangle$; metric $\theta$ (e.g., relative absolute error)

**Output:** Reasoning errors $\eta, \epsilon, \gamma$

**Assumptions:** $\varphi$ composes according to an associative function over the BCCs of causal graph $\mathcal{G}_{XY}$.

    *Compute quantity-wise errors.*
1: **for** all node pairs $\{R_i, L_{j>i}\}$ in $\mathcal{C}_{XY}$ **do**
2:     $\eta_{R_i L_j} \leftarrow \theta(\varphi^*_{R_i L_j}, \widehat{\varphi}_{R_i L_j})$     ▷ External validity.

    *Compute inductive reasoning errors.*
3: **for** all paths $i$ from $X$ to $Y$ in $\mathcal{C}_{XY}$ **do**
4:     Get composition $\widehat{\varphi}^\circ_i$ for path $i$ from knowledge of edges $j \in i$
5:     $\epsilon_i \leftarrow \theta(\varphi^*_{XY}, \widehat{\varphi}^\circ_i)$     ▷ External validity.
6:     $\gamma_i \leftarrow \theta(\widehat{\varphi}_{XY}, \widehat{\varphi}^\circ_i)$     ▷ Internal consistency.
    **return** $\eta, \epsilon, \gamma$

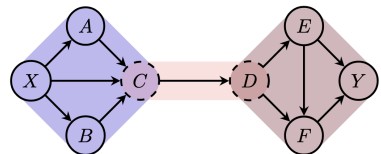

A. ORIGINAL DAG $\mathcal{G}_{XY}$

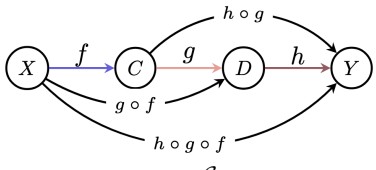

B. CCT $\mathcal{C}_{XY}$

*Figure 4.* A running example for our framework. (**A**) DAG with BCCs (violet, pink, maroon) and cutpoints $C, D$. (**B**) CCT $\mathcal{C}_{XY}$ modeling all commutative CCR paths from $X$ to $Y$. Here, $f := \mathrm{PNS}_{XC}$, $g := \mathrm{PNS}_{CD}$, $h := \mathrm{PNS}_{DY}$, $g \circ f := \mathrm{PNS}_{XD}$, $h \circ g \circ f := \mathrm{PNS}_{XY}$, etc., and composition is multiplicative.

| Global | $\mathrm{PNS}_{XY}$ |
|---|---|
| Local | $\mathrm{PNS}_{XC}, \mathrm{PNS}_{XD}, \mathrm{PNS}_{CD},$ $\mathrm{PNS}_{CY}, \mathrm{PNS}_{DY}$ |
| Composition | $\mathrm{PNS}_{XC}\mathrm{PNS}_{CY}, \mathrm{PNS}_{XD}\mathrm{PNS}_{DY},$ $\mathrm{PNS}_{XC}\mathrm{PNS}_{CD}\mathrm{PNS}_{DY}$ |

*Table 1.* Quantities of interest for inductive CCR over the PNS for Fig. 4. All compositions are equivalent to the global quantity.

the PNS and ATE display a simple compositional form under the following conditions. When the DAG $\mathcal{G}_{XY}$ of a linear SCM satisfies A1–A4, the ATE for the root and leaf of $\mathcal{G}_{XY}$ is a product of the ATE values for the root and leaf of each BCC. This follows from the product-of-coefficients heuristic used in classical path-tracing and mediation analysis (Alwin & Hauser 1975; see Appendix C for a worked example). This is not guaranteed for the ATE in nonlinear data generating processes. Conveniently, we prove in Appendix B.2 that this property extends to the PNS under monotonicity even when causal functions are nonlinear.

**Theorem 5.1** (PNS composition across BCCs)**.** *Given an SCM where DAG $\mathcal{G}_{XY}$ satisfies A1–A4 and $Y$ is monotonic in $X$, the PNS for root $X$ and leaf $Y$ composes as*

$$\mathrm{PNS}_{XY} = \prod_{\{R_i, L_i\} \in \mathbf{C}} \mathrm{PNS}_{R_i L_i} \qquad (13)$$

*where $\mathbf{C}$ is the set of all BCCs in $\mathcal{G}_{XY}$ and $R_i, L_i$ are the root and leaf of BCC $\mathbf{C}_i$, respectively.*

Adjacent BCCs in $\mathcal{G}_{XY}$ can be treated as a single component and Theorem 5.1 still holds (Fig. B.1). In Appendix E, we illustrate Theorem 5.1 with simulations of inductive and deductive CCR for the ATE and PNS. To facilitate evaluation for compositional forms similar to Thm. 5.1, we introduce a new graphical tool for visualizing the flow of causal information through BCCs: the *commutative cut tree*.

**Definition 5.2** (Commutative cut tree (CCT))**.** Let $\mathcal{G}_{XY}$ be a causal graph satisfying A1–A4 and let $\varphi$ be a causal measure that composes according to an associative function over BCCs (e.g., multiplication as in Theorem 5.1). CCT $\mathcal{C}_{XY}$ is a transformation of $\mathcal{G}_{XY}$ that models all CCR pathways from root $X$ to leaf $Y$ for measure $\varphi$. $\mathcal{C}_{XY}$ is obtained by a two-step transformation of $\mathcal{G}_{XY}$:

1. Construct a causal chain with nodes $X \cup \mathbf{S} \cup Y$, where $\mathbf{S}$ is a topological ordering of the cutpoints in $\mathcal{G}_{XY}$ (e.g., $X \to C \to D \to Y$ for Fig. 4A).
2. Add a directed edge between any non-adjacent nodes in the chain to yield a complete graph where all directed paths point from root $X$ to leaf $Y$ (e.g., Fig. 4B).

CCTs abstract away complexity in our original causal graph by collapsing BCCs into single edges. This allows for evaluation on arbitrarily complex DAGs with cutpoints as if they were simply directed chains. This abstraction simplifies the problem representation by (1) marginalizing out variables that are unnecessary for valid causal inference in our setting and (2) visualizing pathways of composition. As demonstrated in Section 5.2, CCTs can be leveraged as a design tool for formulating reasoning tasks. As illustrated in Section 6.2, they can also serve as an interpretable visualization tool for graphically representing CCR errors.

### 5.2. Inductive CCR as Commutative Reasoning

We now propose a means of systematically evaluating CCR that leverages CCTs and Theorem 5.1 (Alg. 1), applicable to the ATE under linearity and the PNS under monotonicity. We can view CCR as reasoning that $\mathcal{C}_{XY}$ is commutative:

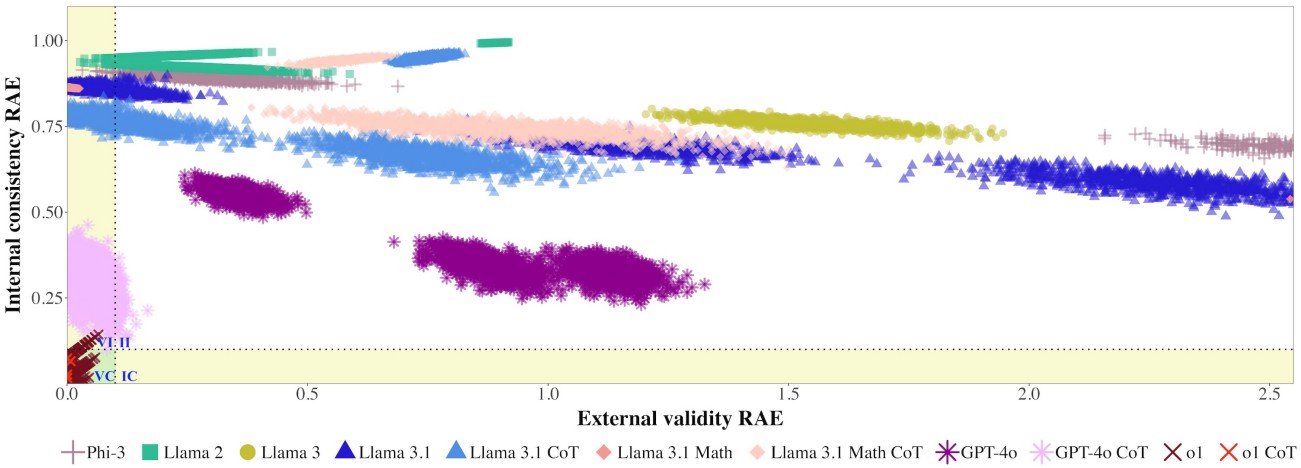

*Figure 5.* For PNS compositions ($n = 1000$ estimates per quantity per model), we compare RAE w.r.t. ground truth (external validity) and $\widehat{\text{PNS}}_{XY}$ (internal consistency) to visualize our four reasoning quadrants (VI/IC in yellow; VC in green; II in white). Dotted lines are error thresholds (RAE = 0.1). Models are listed by increasing size (Table G.1). The *x*-axis is truncated; see Fig. G.11 for the full distribution.

every possible composition (corresponding to the paths from $X$ to $Y$ in $\mathcal{C}_{XY}$) should be equivalent to each other and to ground truth, up to some error. If ground truth values are unavailable, Alg. 1 can be run for internal consistency only. See Appendix D for time complexity.

**Running Example: Intuition for Algorithm 1** For illustration, we walk through Alg. 1 where $\varphi$ is the PNS and DAG $\mathcal{G}_{XY}$ is structured according to Fig. 4A. We continue to employ the notation defined in Section 4. In Section 6, we implement this same walk-through for LM evaluation.

The assumption stated in Alg. 1 is satisfied, as the PNS composes over BCCs by multiplication (Theorem 5.1). To begin, we must determine the quantities of interest for assessing CCR over $\mathcal{G}_{XY}$ (Table 1). To do this, we first obtain the corresponding CCT $\mathcal{C}_{XY}$. The global quantity of interest will always be the PNS for the root and leaf ($\text{PNS}_{XY}$), corresponding to edge $X \rightarrow Y$ in $\mathcal{C}_{XY}$. Local quantities of interest will be the PNS for every remaining pair of nodes in $\mathcal{C}_{XY}$, resulting in $\binom{n}{2} - 1$ quantities (e.g., $\text{PNS}_{CY}$). Finally, we obtain compositions by considering every distinct indirect path from $X$ to $Y$ in $\mathcal{C}_{XY}$. For each such path, the composition of interest is the product of the PNS values for each cause-effect pair on that path. In this case, there are three indirect paths from $X$ to $Y$: $X \rightarrow C \rightarrow Y$, $X \rightarrow D \rightarrow Y$, and $X \rightarrow C \rightarrow D \rightarrow Y$. Thus, our compositions of interest are $\text{PNS}_{XC}\text{PNS}_{CY}$, $\text{PNS}_{XD}\text{PNS}_{DY}$, and $\text{PNS}_{XC}\text{PNS}_{CD}\text{PNS}_{DY}$ (Fig. G.1).

Input to Alg. 1 is the set of estimates $\widehat{\text{PNS}} \cdot = f(\mathcal{A}(\mathcal{Q}_{\text{PNS.}}))$ for each quantity enumerated in Table 1, as well as their ground truth values (if available). Lines 1–2 compute $\theta$ for the external validity of the PNS for each cause-effect pair in $\mathcal{C}_{XY}$. Lines 3–6 compute $\theta$ for the external validity

**Factual prompt**

Xinyu, Ara, Becca, Celine, Daphne, Emma, Fox, and Yasmin are going to a party, where the host is going to distribute candies. Xinyu will be happy if she gets at least 7 candies. Ara will be happy if Xinyu is happy or if he gets at least 7 candies. Becca will be happy if... After distributing the candies, Xinyu gets 4, Ara gets 6, Becca gets 5, Celine gets 10, Daphne gets 1, Emma gets 1, Fox gets 4, and Yasmin gets 3. Is Celine happy? Be as concise as possible.

**Counterfactual prompt**

Now, suppose that Xinyu is happy regardless of the candy distribution. With this assumption, is Celine happy? Be as concise as possible.

*Figure 6.* Factual and counterfactual prompt excerpts. For example, we can obtain $\widehat{\text{PNS}}_{XC}$ by simulating potential outcomes $X = \text{TRUE}$, $X = \text{FALSE}$ (Xinyu is or is not happy) and then querying for the value of $C$ (Celine is or is not happy). Analogously, we obtain $\widehat{\text{PNS}}_{DY}$ with interventions on $D$ (Daphne's happiness) and queries on $Y$ (Yasmin's happiness), etc.

and internal consistency of each composition. Each $\widehat{\varphi}_i^\circ$ corresponds to one distinct path from $X$ to $Y$ in $\mathcal{C}_{XY}$, and is obtained by taking the product of PNS estimates for each cause-effect pair on the path. For example, we estimate $\text{PNS}_{XC}\text{PNS}_{CY}$ as $\widehat{\text{PNS}}_{XC}$ times $\widehat{\text{PNS}}_{CY}$. Finally, Alg. 1 returns all external validity and internal consistency errors.

**Automated Task Generation** We provide open-source code for generating random CCR tasks analogous to our running example. To facilitate future benchmarking, generators allow the user to control aspects of task complexity (e.g., DAG size). See Appendix F for additional functionality.

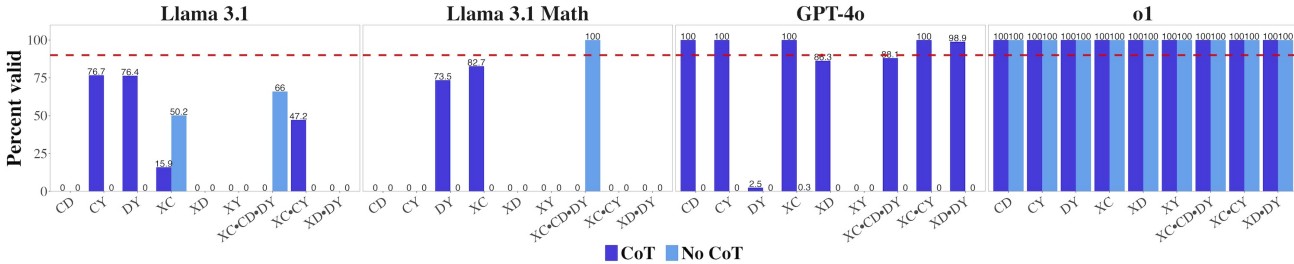

*Figure 7.* Percent of PNS estimates ($n = 1000$) that are externally valid for CoT vs non-CoT prompting. PNS are denoted by cause-effect pair (e.g., $\mathrm{PNS}_{XC}\mathrm{PNS}_{CY}$ as $XC \cdot CY$). Reasoning was externally valid if $\geq 90\%$ of estimates had RAE $\leq 0.1$ (red dashed line).

# 6. Empirical Demonstration in LMs

We demonstrate Alg. 1 for inductive CCR for the PNS with an illustrative math problem graphically represented by Fig. 4. See Appendix G for full experimental details. Experiments were designed to demonstrate the correct implementation of this framework, not to thoroughly benchmark CCR in current LMs. Thus, we focus deeply on one toy problem to highlight the insights that our framework can provide.

## 6.1. Experimental Design

**Models** Inference was performed using seven architectures: Llama 2 (Touvron et al., 2023), 3 (Dubey et al., 2024), 3.1 (Dubey et al., 2024), and 3.1 Math (Toshniwal et al., 2024); Phi-3-Mini (Abdin et al., 2024); GPT-4o (Achiam et al., 2023); and o1 (Jaech et al., 2024) (Table G.1). All LMs were fine-tuned for dialogue. Llama 3.1 Math was also fine-tuned for math, and o1 is a designated reasoning model.

**CCR Task** Let $\mathcal{M}$ be an SCM represented by the DAG in Fig. 4. Variables $\mathbf{V}, \mathbf{U} \in \mathcal{M}$ are binary and causal functions are logical *or* (Eq. G.1). Quantities of interest are in Table 1. We defined CCR task $\mathcal{T} \coloneqq \langle \mathrm{PNS}, \mathcal{M}, \mathcal{Q} \rangle$ where prompts in $\mathcal{Q}$ were based on the `CandyParty` math word problem (Figs. 6, G.2; González & Nori 2024). Approximation errors were obtained by computing relative absolute errors (RAE; Eq. G.3). Thus, we define a task-metric-model triple $\langle \mathcal{T}, \mathrm{RAE}, \mathcal{A} \rangle$ for each model $\mathcal{A}$. Llama 3.1, Llama 3.1 Math, GPT-4o, and o1 were also presented with a CoT wrapper for $\mathcal{Q}$ to assess impacts of prompt formulation on CCR elicitation. The CoT wrapper for our original prompt template presented the model with two worked examples: one factual and one counterfactual (Fig. G.3).

**LMs as Counterfactual Data Simulators** To assess CCR at the counterfactual rung of Pearl's Causal Hierarchy, we treat LMs as *counterfactual data simulators* (González & Nori, 2024; Hüyük et al., 2025). Instead of directly prompting the LM to perform formal causal inference (as in Jin et al. 2023), series of factual and counterfactual "yes/no" questions were submitted to the model (e.g., Figs. 6, G.4, G.5). Natural language responses were then converted to their corresponding Boolean values using Llama 3. The resulting Boolean vectors were treated as samples from observational and interventional distributions, which were then used to compute PNS estimates. For each quantity of interest in $\mathcal{T}$, 1000 PNS estimates were obtained per model. Reasoning was considered externally valid or internally consistent for a quantity if $\geq 90\%$ of the 1000 estimates had RAE $\leq 0.1$ (threshold chosen prior to analysis). Reasoning was deemed near-valid if $\geq 75\%$ of estimates met the threshold.

## 6.2. Experimental Results & Discussion

**General Results** Fig. G.12 plots all PNS distributions. Figs. 5 and G.11 jointly represent internal consistency and external validity RAE for our three compositions of interest, with quadrants representing disparate reasoning profiles: VC, VI, IC, II. Figs. 7, G.13, and G.14 show the percent of PNS estimates that were externally valid for each model. Errors did not decrease monotonically with increasing model size nor Llama version (Figs. G.16, G.17). As demonstrated in Fig. 8, CCTs allow us to see compositional reasoning errors in graphical representation. Only the responses from o1 correctly implied that CCT $\mathcal{C}_{XY}$ was commutative.

**Taxonomy of Reasoners** Results for task $\mathcal{T}$ revealed three taxonomically distinct error patterns: VC (o1), near-VI (GPT-4o with CoT), and II (all others). Only o1 placed mass in the VC reasoning quadrant in Fig. 5. All models besides o1 were $0\%$ valid for global quantity $\mathrm{PNS}_{XY}$, with and without CoT (Figs. 7, G.13, G.14). Without CoT, Phi-3, Llama 2, Llama 3, and GPT-4o failed to exceed $3\%$ validity for any quantity. II reasoners often displayed high variance (Fig. 5). All error distributions for CoT were significantly different from non-CoT, except for o1 on $\mathrm{PNS}_{XY}$ (Wilcoxon test, $\alpha = 0.05$). GPT-4o displayed the greatest sensitivity to prompt formulation, going from II to near-VI. With CoT, $75.1\%$ of estimates from GPT-4o were near-valid and mean external validity on local quantities and compositions exceeded $77\%$ and $95\%$, respectively, but failure on $\mathrm{PNS}_{XY}$ revealed internally inconsistent reasoning.

**CCR Failure Modes** Preliminary error analyses included manual assessment of LM text responses. For Llama mod-

els, poor numeracy explained some errors (Figs. G.4, G.5). GPT-4o with CoT displayed several kinds of faulty reasoning for $\widehat{\text{PNS}}_{XY}$ (Figs. G.7, G.8) and $\widehat{\text{PNS}}_{DY}$ (Figs. G.9, G.10). These included (1) failure to correctly extract causal relations (e.g., true causal parents were missing from the stated parent set; Fig. G.10), (2) incorrect logic despite correct relation extraction (e.g., a clause did not logically follow from the preceding clause; Fig. G.9), and (3) a truncated reasoning process, where the model expressed its logic up to a certain point but ignored the remainder of the underlying causal graph without verbal explanation (e.g., Fig. G.8).

**Errors Increase With Mediation** For all models except GPT-4o with CoT and o1, mean RAE increased monotonically with shortest path distance and total mediating variables between cause and effect (Figs. 9, G.18). Nevertheless, mean RAE for GPT-4o with CoT was $> 10\times$ higher for six mediators than for three mediators. Increasing RAE may be explained by error propagation. Additionally, preliminary analyses suggest that models can "lose the train of thought" along long causal paths (Fig. G.6). These results are in line with prior findings that reasoning performance can diminish with task complexity (Agrawal et al., 2017; Ma et al., 2023; Ray et al., 2023; Press et al., 2023; Jin et al., 2023).

**The Value of Internal Consistency** What does internal consistency tell us that external validity alone cannot? Measuring both reveals not only *if* the reasoner is wrong, but also offers insights into *how* it is wrong. We take VI and IC reasoning as cases in point. First consider GPT-4o with CoT, where external validity alone suggests a degree of successful causal reasoning on task $\mathcal{T}$. However, compositional inconsistencies paint a fuller picture: (1) only two of four paths from $X$ to $Y$ in $\mathcal{C}_{XY}$ commuted (Fig. 8) and (2) GPT-4o was more performant when asked about proximal relationships than distant cause-effect pairs (e.g., $\{X, Y\}$), where the latter required a greater degree of compositional reasoning over indirect relations that were never directly stated in the context prompt (Figs. 9; G.18; G.7–G.10). This failure mode is akin to a student who passes a math quiz by relying more on memorization than true synthesis, and fails to recognize equivalence between multiple formulations of the same problem. Now consider the IC reasoner. This case can arise when a model is *consistently biased*. Say model $\mathcal{A}$ incorrectly infers that $\widehat{\text{PNS}}_{XY} = 2 \times \text{PNS}_{XY}$, but compositions are equally biased: $\widehat{\text{PNS}}_{XC}\widehat{\text{PNS}}_{CY} = 2 \times \text{PNS}_{XY}$, etc. Though these estimates are externally invalid, this overlooks the insight that $\mathcal{A}$ correctly reasoned that the quantities of interest are equivalent. Thus, lumping II with IC reasoners and VI with VC reasoners limits informativeness.

### 6.3. Limitations & Future Directions

This work presents a conceptual foundation for CCR evaluation in LMs. We instantiate this framework for the ATE

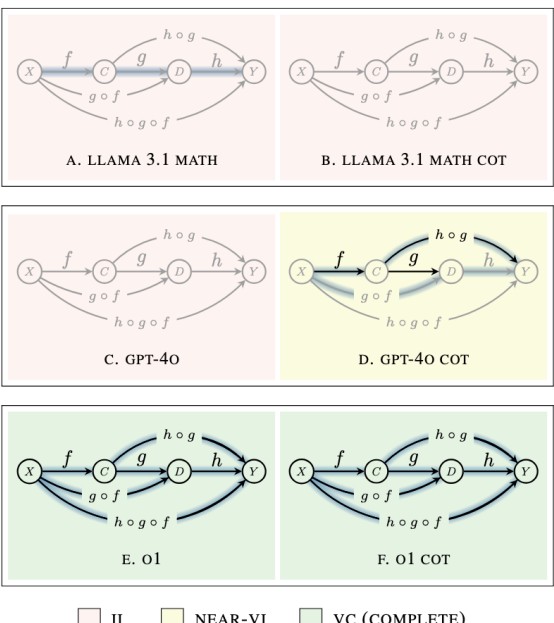

A. LLAMA 3.1 MATH     B. LLAMA 3.1 MATH COT

C. GPT-4O     D. GPT-4O COT

E. O1     F. O1 COT

☐ II    ☐ NEAR-VI    ☐ VC (COMPLETE)

*Figure 8.* Visualizing (in)complete CCR, where only o1 reasons that $\mathcal{C}_{XY}$ commutes. Black edges are externally valid global and local quantities, while gray are invalid. Each path from $X$ to $Y$ that is highlighted in blue represents an externally valid composition. Nodes are black when all paths passing through them are valid.

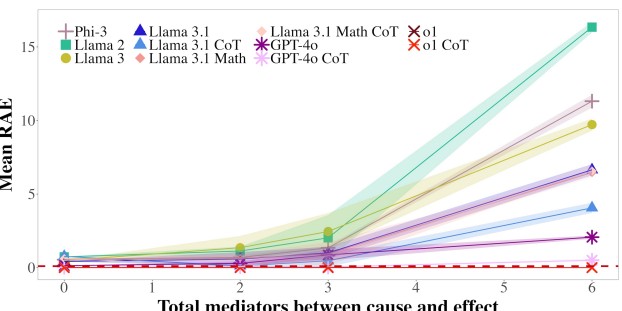

*Figure 9.* Red dashed line denotes the external validity cutoff (RAE = 0.1), with standard deviations in shaded regions.

and PNS in causal graphs with cutpoints (Alg. 1). Empirics are limited to one illustrative task as proof of viability. Results demonstrate how this framework can provide a richer picture of causal reasoning than measuring quantity-wise external validity alone, as is conventionally done. Complete reasoning by o1 validates the correctness of our implementation, though future work should emphasize significantly greater task complexity. Open source and smaller models did notably worse on this task than GPT-4o with CoT and o1, suggesting that even low-complexity CCR is an area on which significant progress can still be made for an important class of LMs (smaller and/or open). As this work only considers the ATE and PNS under Theorem 5.1, extensions could consider other estimands and compositional forms.

## Impact Statement

The potential for reasoning emergence in AI has broad scientific, economic, and social implications for global society. These implications include matters of safety and fairness. This work aims to promote the rigorous measurement of reasoning behaviors in LMs. Success on CCR tasks such as those described in this work is *necessary but not sufficient* for demonstrating that LMs can reason. We encourage cautious interpretation of results, particularly with respect to claims of reasoning in models that are deployed in safety-critical contexts.

## Acknowledgments

Author J. Maasch acknowledges the US National Science Foundation Graduate Research Fellowship under Grant No. DGE–2139899.

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

# APPENDIX

## A. EXTENDED BACKGROUND

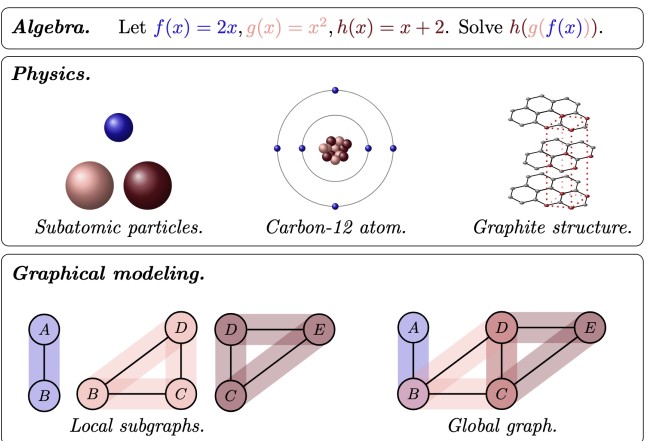

*Figure A.1.* Composition is ubiquitous in physical and symbolic systems, operating at multiple scales of granularity: atoms → molecules → materials; cells → tissues → organs; words → sentences; subroutines → programs; graphical substructures → global graphs; etc.

**Compositional Reasoning in AI** Compositional reasoning has been explored for diverse tasks in AI. These include problem-solving in math, logic, and programming (Saxton et al., 2019; Dziri et al., 2023; Lu et al., 2023a); image generation (Du et al., 2023; Okawa et al., 2023), decomposition (Su et al., 2024), and captioning (Shi et al., 2021); visual concept learning (Wüst et al., 2024); object recognition (Stone et al., 2017); representation learning (Xu et al., 2022); chain-of-thought (Li et al., 2023); visual question answering (Agrawal et al., 2017); and text-to-image retrieval (Ray et al., 2023). Models have been shown to struggle with compositional reasoning in vision and language, especially for complex compositions (Agrawal et al., 2017; Ma et al., 2023; Ray et al., 2023). Saxton et al. (2019) address compositional math reasoning with an emphasis on compositions of functions and manipulation of intermediate results, though causal measures are not explored. Many works center on the development of benchmark datasets for vision and/or language models (e.g., Johnson et al. 2017; Agrawal et al. 2017; Suhr et al. 2017; 2018; Thrush et al. 2022; Ma et al. 2023; Hsieh et al. 2023; Ray et al. 2023). This work introduces new core concepts in CCR evaluation for LMs, on which future benchmarks can be based.

**Applications of the PrC in AI** The majority of works on the PrC have centered on identifiability and the derivation of bounds under various conditions (Tian & Pearl, 2000; Mueller et al., 2022; Sani et al., 2023; Sani & Mastakouri, 2023; Li & Pearl, 2024; Maiti et al., 2025). The PrC have found applications in representation learning (Wang & Jordan, 2022), causal discovery (Cai et al., 2023), and explainable AI and rationalization (Zhang et al., 2023). In LMs, the PrC have been used to evaluate causal reasoning (González & Nori, 2024) and to facilitate fine-tuning for counterfactual reasoning (Hüyük et al., 2025). To our knowledge, ours is the first work to explore the utility of PrC compositionality for measuring compositional reasoning in AI.

**Commutative Diagrams in Reasoning Evaluation** Classically, commutative diagrams have been used as tools for theorem proving (see *diagram chasing*, De Toffoli 2017). In this work, we introduce a special form of commutative diagram for modeling reasoning. González & Nori (2024) use commutative diagrams for causal reasoning evaluation in LMs, where one pathway corresponds to the true problem solution for a math problem and the other corresponds to the reasoning pathway of the LM. The present work shows how commutative diagrams in the form of CCTs can provide compact and exhaustive representations for (1) visualizing CCR pathways and (2) systematizing CCR evaluation.

**Causal Reasoning Evaluation** See Table A.1 for an extensive comparison to prior works in this area. To further clarify our contributions, we compare to the CLADDER benchmark for formal causal reasoning (Jin et al., 2023). Most prior causal reasoning evaluation centers on commonsense knowledge and relation extraction (Table A.1). Like CLADDER, our framework and task generator can be used to (1) probe reasoning at the associational, interventional, and counterfactual

levels; (2) apply formal causal inference methods; and (3) evaluate causal relation extraction (implicitly or explicitly). Our method notably departs from CLADDER in the following ways: (1) we evaluate compositional reasoning, another dimension of causal reasoning that has received little attention in LMs; (2) while CLADDER uses simple graphs (3–4 nodes), our task generator allows the user to freely scale the total number of nodes and our empirical results demonstrate on a graph with eight nodes; (3) we introduce new metrics for the internal consistency of causal reasoning; and (4) we do not require ground truth quantities, as the user can opt to evaluate on internal consistency only. While CLADDER prompts the model to perform causal inference within a single text response, the implementation demonstrated here takes an alternative approach: series of questions are posed to the model, and responses are treated as samples from observational and interventional data distributions. These samples are then used to perform formal causal inference, with the resulting counterfactual quantities (e.g., the PNS) representing the causal inferences *implied* by the model's reasoning at the associational, interventional, and counterfactual levels. Thus, the current implementation applies formal causal inference, but does not directly ask the LM to formalize the causal query (as in Jin et al. 2023). However, the approach proposed by Jin et al. (2023) could be adapted to measure CCR under alternative prompt formulations.

| | QUESTIONS | | | SKILLS | | | | | |
|---|---|---|---|---|---|---|---|---|---|
| | R1 | R2 | R3 | CI | Formal | CRE | QR | Comp | Scale |
| ***Causality as Knowledge (Commonsense)*** | | | | | | | | | |
| COPA (Gordon et al., 2012) | ✗ | ✓ | ✗ | ✗ | ✗ | ✗ | ✗ | ✗ | ✗ |
| Event2Mind (Rashkin et al., 2018) | ✗ | ✓ | ✗ | ✗ | ✗ | ✗ | ✗ | ✗ | ✗ |
| ATOMIC (Sap et al., 2019a) | ✗ | ✓ | ✗ | ✗ | ✗ | ✗ | ✗ | ✗ | ✗ |
| SocialIQA (Sap et al., 2019b) | ✗ | ✓ | ✗ | ✗ | ✗ | ✗ | ✗ | ✗ | ✗ |
| TimeTravel (Qin et al., 2019) | ✗ | ✓ | ✗ | ✗ | ✗ | ✗ | ✗ | ✗ | ✗ |
| Goal-Step (Zhang et al., 2020) | ✗ | ✓ | ✗ | ✗ | ✗ | ✗ | ✗ | ✗ | ✗ |
| Abductive (ART) (Bhagavatula et al., 2020) | ✗ | ✓ | ✗ | ✗ | ✗ | ✗ | ✗ | ✗ | ✗ |
| Com2Sense (Singh et al., 2021) | ✗ | ✓ | ✗ | ✗ | ✗ | ✗ | ✗ | ✗ | ✗ |
| CRASS (Frohberg & Binder, 2022) | ✗ | ✗ | ✓ | ✗ | ✗ | ✗ | ✗ | ✗ | ✗ |
| ***Causality as Language Comprehension (CRE)*** | | | | | | | | | |
| SemEval2021 Task8 (Hendrickx et al., 2010) | ✗ | ✗ | ✗ | ✗ | ✗ | ✓ | ✗ | ✗ | ✗ |
| EventCausality (Do et al., 2011) | ✗ | ✗ | ✗ | ✗ | ✗ | ✓ | ✗ | ✗ | ✗ |
| Causal-TimeBank (Mirza et al., 2014) | ✗ | ✗ | ✗ | ✗ | ✗ | ✓ | ✗ | ✗ | ✗ |
| CaTeRS (Mostafazadeh et al., 2016) | ✗ | ✗ | ✗ | ✗ | ✗ | ✓ | ✗ | ✗ | ✗ |
| BECauSE (Dunietz et al., 2017) | ✗ | ✗ | ✗ | ✗ | ✗ | ✓ | ✗ | ✗ | ✗ |
| TellMeWhy (Lal et al., 2021) | ✗ | ✗ | ✗ | ✗ | ✗ | ✓ | ✗ | ✗ | ✗ |
| ***Formal Causal Reasoning*** | | | | | | | | | |
| Corr2Cause (Jin et al., 2024) | ✗ | ✓ | ✗ | ✓ | ✓ | ✗ | ✗ | ✗ | ✓ |
| CLADDER (Jin et al., 2023) | ✓ | ✓ | ✓ | ✓ | ✓ | ✓ | ✓ | ✗ | ✗ |
| CCR framework and task generator (ours) | ✓ | ✓ | ✓ | ✓ | ✗ | ✓ | ✓ | ✓ | ✓ |

*Table A.1.* Comparison of causal reasoning evaluation frameworks for LMs (adapted directly from Table 9 in Jin et al. 2023). Question types concern the three rungs of Pearl's Causal Hierarchy (Bareinboim et al., 2022): associational (*R1*), interventional (*R2*), and counterfactual (*R3*). Skill types entail the application of causal inference methods (*CI*), formalization of causal queries (*Formal*), causal relation extraction from text (*CRE*), qualitative reasoning (*QR*), systematic composition of causal measures (*Comp*), and scaling the graphical complexity of the causal DAG (*Scale*). In our automated task generator (https://jmaasch.github.io/ccr/), we implement causal prompt templates that require either quantitative reasoning (i.e., numeracy) or qualitative reasoning (e.g., reasoning over colors instead of numbers). Note that we compare to CCR at the conceptual framework level and the task instantiation level. Unlike Corr2Cause and CLADDER, the instantiation presented in Section 6 requires the responses to multiple prompts to be aggregated, which somewhat complicates comparison. While Corr2Cause and CLADDER prompt for causal query formalization and application of causal inference methods *directly* within a single LM response, the experiments in this work treat the LM as a *counterfactual data simulator* whose responses about interventional outcomes logically imply the formal counterfactual quantity, which is then formally computed using the vector of LM responses. However, our conceptual framework (as presented in Section 4) is fully compatible with the direct strategy under alternative prompt formulations, such as application of CLADDER's CausalCoT (an area for future work).

# B. PROBABILITIES OF CAUSATION: COMPOSITIONALITY

## B.1. PN & PS

The PS considers the *presence* of a causal process that can produce a given effect, while the PN considers the *absence* of alternative explanatory processes (Tian & Pearl, 2000). Like the PNS, the PN and PS are point identifiable when $Y$ is monotonic in $X$.

**Definition B.1** (Probability of Necessity (PN), Pearl 1999). The probability that event $y$ would *not* have occurred in the absence of event $x$, given that $x$ and $y$ did jointly occur, is given as

$$PN := \mathbb{P}(y'_{x'} \mid x, y) \tag{B.1}$$
$$= \mathbb{P}(Y_{x'} = \text{FALSE} \mid X = \text{TRUE}, Y = \text{TRUE}).$$

**Definition B.2** (Probability of Sufficiency (PS), Pearl 1999). The probability that event $x$ would produce event $y$ given that $x$ and $y$ did *not* in fact occur is given as

$$PS := \mathbb{P}(y_x \mid x', y'). \tag{B.2}$$

**Definition B.3** (Point identification of the PN and PS under monotonicity, Tian & Pearl 2000). Given a positive-Markovian SCM for which $Y$ is monotonic in $X$ and the causal effects $\mathbb{P}(y_x)$ and $\mathbb{P}(y_{x'})$ are identifiable by Equation 1, the probabilities of causation are point identifiable by the following expressions.

$$PN = \frac{\mathbb{P}(y) - \mathbb{P}(y_{x'})}{\mathbb{P}(x, y)} = \frac{\mathbb{P}(y) - \mathbb{P}(y \mid do(x'))}{\mathbb{P}(x, y)} \tag{B.3}$$

$$PS = \frac{\mathbb{P}(y_x) - \mathbb{P}(y)}{\mathbb{P}(x', y')} = \frac{\mathbb{P}(y \mid do(x)) - \mathbb{P}(y)}{\mathbb{P}(x', y')}. \tag{B.4}$$

## B.2. PrC Compositionality: Proofs

*Proof.* Here we show that the PN and PS do not compose according to Theorem 5.1, while the PNS does. Consider a causal model $X \to Y \to Z$ with binary variables $X, Y, Z$. Assume that all relations in the causal graph are monotonic such that

$$\mathbb{P}(y'_x, y_{x'}) = 0 \tag{B.5}$$
$$\mathbb{P}(z'_y, z_{y'}) = 0. \tag{B.6}$$

We start by expressing $PN_{XY}, PS_{XY}$ in terms of probabilities over potential outcomes $Y_x, Y_{x'}$:

$$PN_{XY} = \mathbb{P}(y'_{x'}|x, y) = \frac{\mathbb{P}(y, y'_{x'}|x)}{\mathbb{P}(y, y_{x'}|x) + \mathbb{P}(y, y'_{x'}|x)} = \frac{\mathbb{P}(y_x, y'_{x'})}{\mathbb{P}(y_x, y_{x'}) + \mathbb{P}(y_x, y'_{x'})} \tag{B.7}$$

$$PS_{XY} = \mathbb{P}(y_x|x', y') = \frac{\mathbb{P}(y_x, y'|x')}{\mathbb{P}(y_x, y'|x') + \mathbb{P}(y'_x, y'|x')} = \frac{\mathbb{P}(y_x, y'_{x'})}{\mathbb{P}(y_x, y'_{x'}) + \mathbb{P}(y'_x, y'_{x'})}. \tag{B.8}$$

Using these expressions, together with the monotonicity assumption $\mathbb{P}(y'_x, y_{x'}) = 0$ and the simple fact that

$$\mathbb{P}(y_x, y_{x'}) + \mathbb{P}(y_x, y'_{x'}) + \mathbb{P}(y'_x, y'_{x'}) + \mathbb{P}(y'_x, y_{x'}) = 1, \tag{B.9}$$

we can fully describe the distribution of potential outcomes $Y_x, Y_{x'}$ (by solving the resulting system of equations):

$$\mathbb{P}(y_x, y_{x'}) = \frac{1/PN_{XY} - 1}{1/PN_{XY} + 1/PS_{XY} - 1} \tag{B.10}$$

$$\mathbb{P}(y_x, y'_{x'}) = \frac{1}{1/PN_{XY} + 1/PS_{XY} - 1} \tag{B.11}$$

$$\mathbb{P}(y'_x, y'_{x'}) = \frac{1/PS_{XY} - 1}{1/PN_{XY} + 1/PS_{XY} - 1} \tag{B.12}$$

$$\mathbb{P}(y'_x, y_{x'}) = 0. \tag{B.13}$$

Similarly, given $PN_{YZ}, PS_{YZ}$, we describe the distribution of potential outcomes $Z_y, Z_{y'}$:

$$\mathbb{P}(z_y, z_{y'}) = \frac{1/PN_{YZ} - 1}{1/PN_{YZ} + 1/PS_{YZ} - 1} \tag{B.14}$$

$$\mathbb{P}(z_y, z'_{y'}) = \frac{1}{1/PN_{YZ} + 1/PS_{YZ} - 1} \tag{B.15}$$

$$\mathbb{P}(z'_y, z'_{y'}) = \frac{1/PS_{YZ} - 1}{1/PN_{YZ} + 1/PS_{YZ} - 1} \tag{B.16}$$

$$\mathbb{P}(z'_y, z_{y'}) = 0. \tag{B.17}$$

Since we have now fully characterized our causal model, we can easily derive formulas for $PN_{XZ}$ or $PS_{XZ}$. Starting with $PN_{XZ}$, we first observe that

$$PN_{XZ} = \frac{\mathbb{P}(z_x, z'_{x'})}{\mathbb{P}(z_x, z_{x'}) + \mathbb{P}(z_x, z'_{x'})}, \tag{B.18}$$

as we have done with $PN_{XY}$ earlier. Then, we note that

$$\mathbb{P}(z_x, z'_{x'}) = \mathbb{P}(y_x, y_{x'})\mathbb{P}(z_x, z'_{x'}|y_x, y_{x'}) + \mathbb{P}(y_x, y'_{x'})\mathbb{P}(z_x, z'_{x'}|y_x, y'_{x'}) + \mathbb{P}(y'_x, y'_{x'})\mathbb{P}(z_x, z'_{x'}|y'_x, y'_{x'}) \tag{B.19}$$

$$= \mathbb{P}(y_x, y_{x'})\mathbb{P}(z_y, z'_y) + \mathbb{P}(y_x, y'_{x'})\mathbb{P}(z_y, z'_{y'}) + \mathbb{P}(y'_x, y'_{x'})\mathbb{P}(z_{y'}, z'_{y'}) \tag{B.20}$$

$$= \mathbb{P}(y_x, y'_{x'})\mathbb{P}(z_y, z'_{y'}) \tag{B.21}$$

$$= \frac{1}{1/PN_{XY} + 1/PS_{XY} - 1} \times \frac{1}{1/PN_{YZ} + 1/PS_{YZ} - 1} \tag{B.22}$$

Note that we obtain Equation B.21 from B.20 by monotonicity: $\mathbb{P}(y'_x) = 0$ and $\mathbb{P}(z'_y) = 0$ (as outcome cannot be false when treatment is true). Then,

$$\mathbb{P}(z_x, z_{x'}) = \mathbb{P}(y_x, y_{x'})\mathbb{P}(z_x, z_{x'}|y_x, y_{x'}) + \mathbb{P}(y_x, y'_{x'})\mathbb{P}(z_x, z_{x'}|y_x, y'_{x'}) + \mathbb{P}(y'_x, y'_{x'})\mathbb{P}(z_x, z_{x'}|y'_x, y'_{x'}) \tag{B.23}$$

$$= \mathbb{P}(y_x, y_{x'})\mathbb{P}(z_y, z_y) + \mathbb{P}(y_x, y'_{x'})\mathbb{P}(z_y, z_{y'}) + \mathbb{P}(y'_x, y'_{x'})\mathbb{P}(z_{y'}, z_{y'}) \tag{B.24}$$

$$= \mathbb{P}(y_x, y_{x'})\mathbb{P}(z_y) + \mathbb{P}(y_x, y'_{x'})\mathbb{P}(z_y, z_{y'}) + \mathbb{P}(y'_x, y'_{x'})\mathbb{P}(z_{y'}) \tag{B.25}$$

$$= \mathbb{P}(y_x, y_{x'})(\mathbb{P}(z_y, z_{y'}) + \mathbb{P}(z_y, z'_{y'})) + \mathbb{P}(y_x, y'_{x'})\mathbb{P}(z_y, z_{y'}) + \mathbb{P}(y'_x, y'_{x'})(\mathbb{P}(z_y, z_{y'}) + \mathbb{P}(z'_y, z_{y'})) \tag{B.26}$$

$$= \frac{1}{1/PN_{XY} + 1/PS_{XY} - 1} \times \frac{1}{1/PN_{YZ} + 1/PS_{YZ} - 1} \times$$
$$\left[ \left( \frac{1}{PN_{XY}} - 1 \right) \left( \frac{1}{PN_{YZ}} \right) + \left( \frac{1}{PN_{YZ}} - 1 \right) + \left( \frac{1}{PS_{XY}} - 1 \right) \left( \frac{1}{PN_{YZ}} - 1 \right) \right]. \tag{B.27}$$

Therefore,

$$\frac{1}{PN_{XZ}} = \frac{1}{PN_{XY}} \frac{1}{PN_{YZ}} + \left( \frac{1}{PS_{XY}} - 1 \right) \left( \frac{1}{PN_{YZ}} - 1 \right), \tag{B.28}$$

which cannot be reduced further to a function of $PN_{XY}$ and $PN_{YZ}$ only. This is because our causal model has enough degrees of freedom to fix $PN_{XY}$ and $PN_{YZ}$ but still vary $PS_{XY}$ resulting in different values of $PN_{XZ}$.

Following a similar derivation, we can also write

$$\frac{1}{PS_{XZ}} = \frac{1}{PS_{XY}} \frac{1}{PS_{YZ}} + \left( \frac{1}{PN_{XY}} - 1 \right) \left( \frac{1}{PS_{YZ}} - 1 \right). \tag{B.29}$$

**Thus, while individual $PN$ and $PS$ values compose in a rather complicated way, $PNS$ values happen to compose multiplicatively.** We have already proven this statement along the way in (B.21):

$$PNS_{XZ} = \mathbb{P}(z_x, z'_{x'}) = \mathbb{P}(y_x, y'_{x'})\mathbb{P}(z_y, z'_{y'}) = PNS_{XY} PNS_{YZ} \tag{B.30}$$

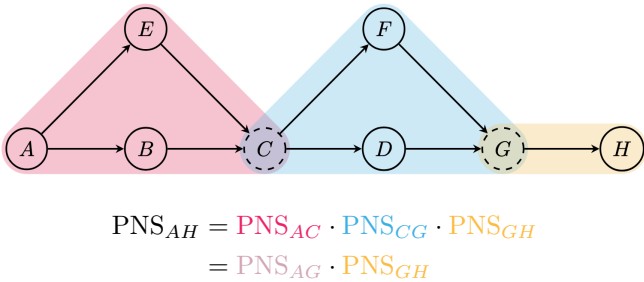

$$\text{PNS}_{AH} = \text{PNS}_{AC} \cdot \text{PNS}_{CG} \cdot \text{PNS}_{GH}$$
$$= \text{PNS}_{AG} \cdot \text{PNS}_{GH}$$

*Figure B.1.* Multiplicative composition of the PNS across BCCs when cause-effect pairs satisfy the monotonocity constraint. Nodes with dashed outlines are cutpoints.

**Thus, the PNS composes according to Theorem 5.1 under monotonicity.** To obtain (B.21), we used monotonicity in (B.19). Otherwise, there would have been a fourth term in (B.19) related to $\mathbb{P}(y'_x, y_{x'})$ and we would have been left with

$$\mathbb{P}(z_x, z'_{x'}) = \mathbb{P}(y_x, y'_{x'})\mathbb{P}(z_y, z'_{y'}) + \mathbb{P}(y'_x, y_{x'})\mathbb{P}(z'_y, z_{y'}). \tag{B.31}$$

This is intuitive: if $X \iff Z$, then either $X \iff Y \wedge Y \iff Z$ or $X \iff Y' \wedge Y' \iff Z$. Then, you can argue that $PNS_{XZ}$ cannot be a pure function of $PNS_{XY}$ and $PNS_{YZ}$: we can fix $PNS_{XY} = \mathbb{P}(y_x, y'_{x'})$ and $PNS_{YZ} = \mathbb{P}(z_y, z'_{y'})$ and still easily vary $\mathbb{P}(y'_x, y_{x'})$ or $\mathbb{P}(z'_y, z_{y'})$ resulting in different values of $PNS_{XZ}$.

$\square$

## B.3. Extended Discussion of the PNS

### B.3.1. THE PNS COINCIDES WITH THE ATE UNDER MONOTONICITY

We highlight the relationship of the PNS to another causal quantity: the *average treatment effect* (ATE). Under monotonicity, the PNS is identifiable from causal effects. In the binary setting, the causal effect $\mathbb{E}[Y \mid do(X = x)]$ can be expressed as

$$\mathbb{E}[Y \mid do(X = x)] = 1 \cdot \mathbb{P}(y = 1 \mid do(X = x)) + 0 \cdot \mathbb{P}(y = 0 \mid do(X = x)) \tag{B.32}$$
$$= \mathbb{P}(y = 1 \mid do(X = x)). \tag{B.33}$$

**Proposition B.4.** *Following from Eq. B.33, we can express the ATE as a difference of probabilities that is equal to the PNS.*

$$\text{ATE} \coloneqq \mathbb{E}[Y \mid do(X = x)] - \mathbb{E}[Y \mid do(X = x')] \tag{B.34}$$
$$= \mathbb{P}(y \mid do(X = x)) - \mathbb{P}(y \mid do(X = x')) \tag{B.35}$$
$$= \text{PNS}. \tag{B.36}$$

# C. ATE COMPOSITION IN LINEAR SCMs

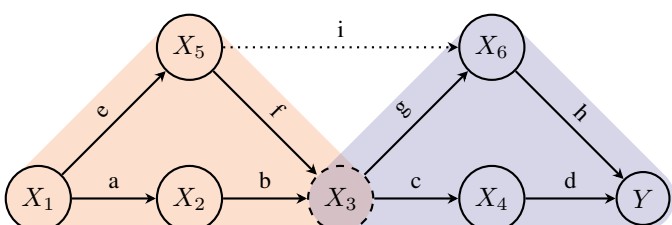

*Figure C.1.* DAG $\mathcal{G}_{X_1 Y}$ contains a subgraph that can be decomposed into two BCCs sharing cutpoint $X_3$. $\mathcal{G}_{X_1 X_3}$ is the BCC induced by edges in orange, $\mathcal{G}_{X_3 Y}$ by edges in periwinkle. Assume a linear SCM. If the dotted edge from $X_5$ to $X_6$ does not exist, $\text{ATE}_{X_1 Y} = \text{ATE}_{X_1 X_3} \cdot \text{ATE}_{X_3 Y}$. If the dotted edge does exist, then this product is summed with an additional term corresponding to the path-specific effect for path $X_1 \to X_5 \to X_6 \to Y$, which does not pass through $X_3$.

## C.1. Worked Example

In linear SCMs, composition of the ATE over the BCCs of a DAG shares the same form as Theorem 5.1. This follows from the product-of-coefficients heuristic used in classical path-tracing and linear mediation analysis (Wright, 1921; Alwin & Hauser, 1975; MacKinnon, 2012; Imai et al., 2010; Pearl, 2012; 2013; Singal & Michailidis, 2024). Linear path-tracing rules are not guaranteed to apply in nonlinear data generating processes, as addressed in nonparametric causal mediation analysis (Pearl, 2001; 2012) and nonparametric path-tracing (Zhang & Bareinboim, 2018).

We provide an illustrative example of composition for the ATE in linear SCMs for two settings: (1) when the DAG contains at least one cutpoint and (2) when the DAG does not contain a cutpoint, but does contain a subgraph with at least one cutpoint. Finite sample results are in Figure C.2.

**DAGs With Cutpoints**  Take Figure C.1 as an example. First, we assume that the dotted edge from $X_5$ to $X_6$ does not exist. In this case, the DAG contains cutpoint $X_3$ and two BCCs. The ATE for $\{X_1, Y\}$ can then be expressed as a product of the ATE values corresponding to the root and leaf of each BCC.

$$\text{ATE}_{X_1 X_3} = ab + ef \tag{C.1}$$
$$\text{ATE}_{X_3 Y} = cd + gh \tag{C.2}$$
$$\text{ATE}_{X_1 Y} = abcd + abgh + efcd + efgh \tag{C.3}$$
$$\text{ATE}_{X_1 X_3} \cdot \text{ATE}_{X_3, Y} = (ab + ef)(cd + gh) = \text{ATE}_{X_1 Y} \tag{C.4}$$

We can see that the ground truth $\text{ATE}_{X_1, Y}$ expressed in Equation C.3 is equivalent to the product expressed in Equation C.4.

**DAGs Without Cutpoints**  Next, assume that the edge from $X_5$ to $X_6$ does exist.

$$\text{ATE}_{X_1 Y} = abcd + abgh + efcd + efgh + eih \tag{C.5}$$
$$= (ab + ef)(cd + gh) + eih \tag{C.6}$$
$$= \text{ATE}_{X_1 X_3} \cdot \text{ATE}_{X_3 Y} + \text{PSE} \tag{C.7}$$

where PSE is the path-specific effect for $X_1 \to X_5 \to X_6 \to Y$, the only causal path for $\{X_1, Y\}$ not passing through cut vertex $X_3$.

**Finite Sample Simulation**  In Figure C.2, we demonstrate finite sample results for the case where $X_5 \to X_6$ and $X_5 \not\to X_6$. Exogenous variables are drawn from the standard normal distribution. All edge coefficients in Figure C.1 are set to 1.5. Thus, we have the following true ATE values when $X_5 \not\to X_6$:

$$\text{ATE}_{X_1 3} = 1.5^2 + 1.5^2 = 4.5 \tag{C.8}$$
$$\text{ATE}_{X_3 Y} = 1.5^2 + 1.5^2 = 4.5 \tag{C.9}$$
$$\text{ATE}_{X_1 Y} = 4(1.5^4) = 20.25. \tag{C.10}$$

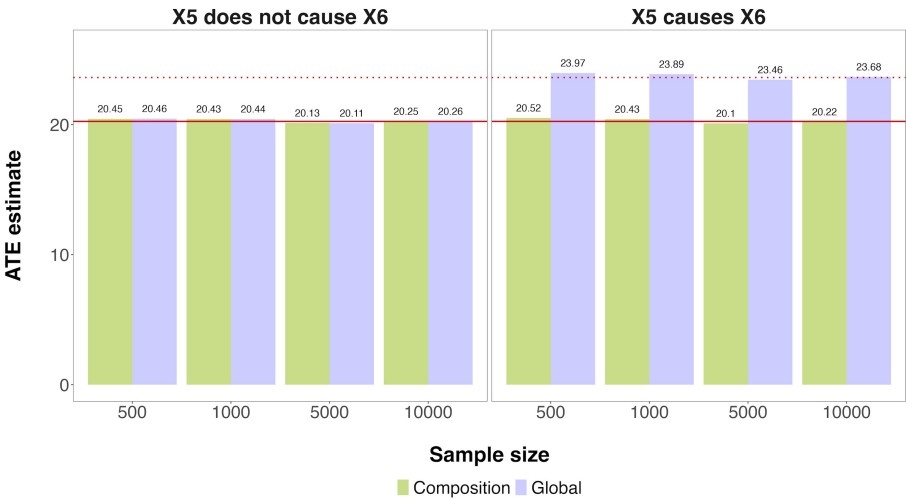

*Figure C.2.* ATE composition for a linear SCM whose graphical representation is given by Figure C.1. Composition $= \text{ATE}_{X_1 X_3} \cdot \text{ATE}_{X_3 Y}$ and global $= \text{ATE}_{X_1 Y}$. The solid red line represents the true $\text{ATE}_{X_1 Y}$ when $X_5 \nrightarrow X_6$, and the dotted red line represents $\text{ATE}_{X_1 Y}$ when $X_5 \rightarrow X_6$. Estimates were obtained by linear regression (https://scikit-learn.org/).

When $X_5 \rightarrow X_6$, we have the additional path-specific effect for $X_1 \rightarrow X_5 \rightarrow X_6 \rightarrow Y$, which is equal to $1.5^3 = 3.375$. Thus,

$$\text{ATE}_{X_1 Y} = 20.25 + 3.375 = 23.625. \tag{C.11}$$

As shown in Figure C.2, finite sample results approach the true values with small errors. Note that estimation of $\text{ATE}_{X_3 Y}$ when $X_5 \rightarrow X_6$ requires covariate adjustment for $X_5$, which acts as a confounder for $X_3, Y$ in this case.

# D. ALGORITHM 1: INDUCTIVE CCR EVALUATION

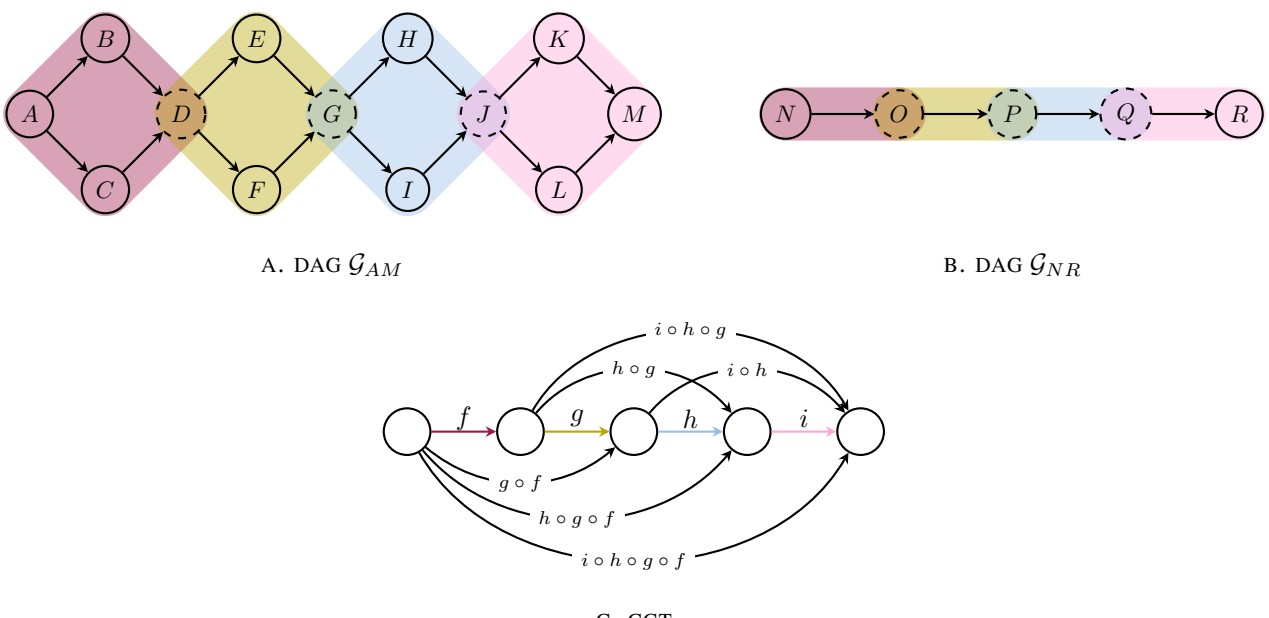

A. DAG $\mathcal{G}_{AM}$

B. DAG $\mathcal{G}_{NR}$

C. CCT

*Figure D.1.* (**A**) DAG $\mathcal{G}_{AM}$, whose undirected skeleton is a cactus graph with three cutpoints (dashed nodes). For $\mathcal{G}_{AM}$, the global PNS $\text{PNS}_{AM}$ is equivalent to the product $\text{PNS}_{AD}\text{PNS}_{DG}\text{PNS}_{GJ}\text{PNS}_{JM}$, while $\text{PNS}_{AG}$ decomposes as $\text{PNS}_{AD}\text{PNS}_{DG}$, etc. (**B**) DAG $\mathcal{G}_{NR}$, a directed chain. (**C**) The CCT shared by both $\mathcal{G}_{AM}$ and $\mathcal{G}_{NR}$, which models all CCR pathways from root to leaf.

## D.1. Time Complexity of Algorithm 1

Let $n$ be the number of nodes in the original causal DAG. In the worst case, the number of nodes in the corresponding CCT will be $n$ (e.g., Figure D.1B). The first for-loop in Algorithm 1 (Lines 2-3) requires errors to be computed for every pair of nodes in the CCT, resulting in $\binom{n}{2}$ errors. Thus, the first for-loop requires the calculation of $O(n^2)$ errors.

Let $k$ be the number of unique paths from root to leaf in the CCT. The second for-loop in Algorithm 1 (Lines 3-6) requires two errors to be computed for all $(k-1)$ compositions: one for internal consistency and one for external validity. The total number of unique paths from root to leaf is $k = 2^{n-2}$, where $n-2$ is the number of cutpoints in the original DAG. Thus, $O(2^n)$ errors must be calculated.

Therefore, the total number of errors to be calculated will be $O(2^n)$. In Table D.1 and Figures D.2 and D.3, we empirically demonstrate that the total number of unique paths from root to leaf in a CCT is $2^{n-2}$.

| NODES ($n$) | CUTPOINTS ($n-2$) | EDGES $\binom{n}{2}$ | PATHS ($2^{n-2}$) |
|---|---|---|---|
| 3 | 1 | 3 | 2 |
| 4 | 2 | 6 | 4 |
| 5 | 3 | 10 | 8 |
| 6 | 4 | 15 | 16 |
| 7 | 5 | 21 | 32 |
| 8 | 6 | 28 | 64 |
| 9 | 7 | 36 | 128 |
| 10 | 8 | 45 | 256 |
| 11 | 9 | 55 | 512 |

*Table D.1.* Total cutpoints, edges, and unique paths from root to leaf in CCTs as total nodes scales.

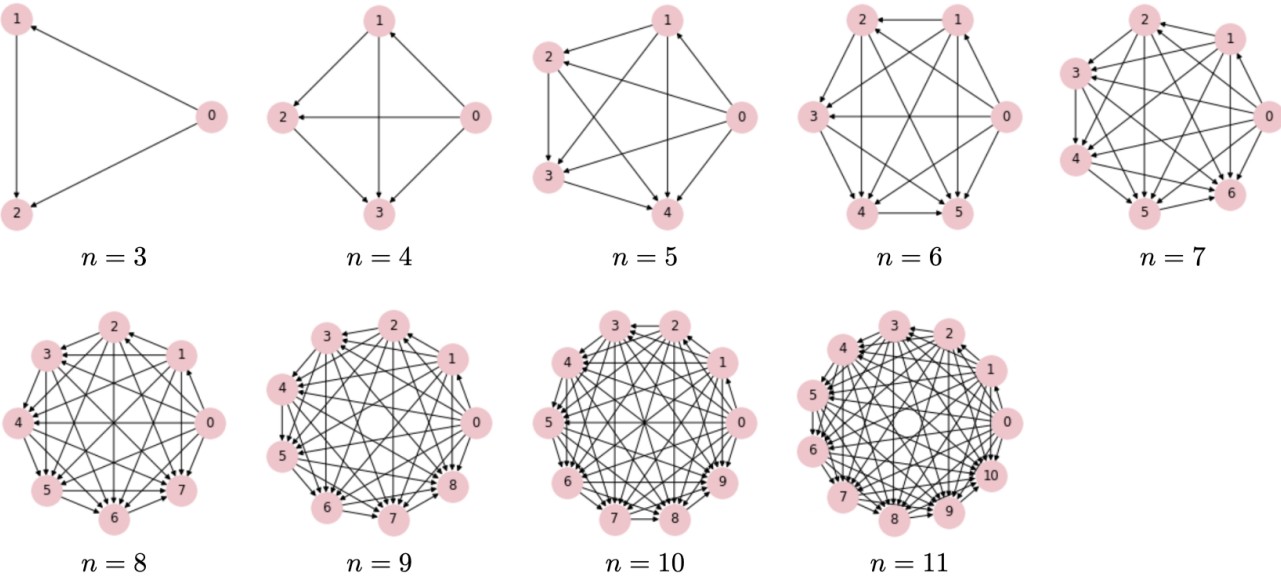

*Figure D.2.* CCTs with $n$ nodes.

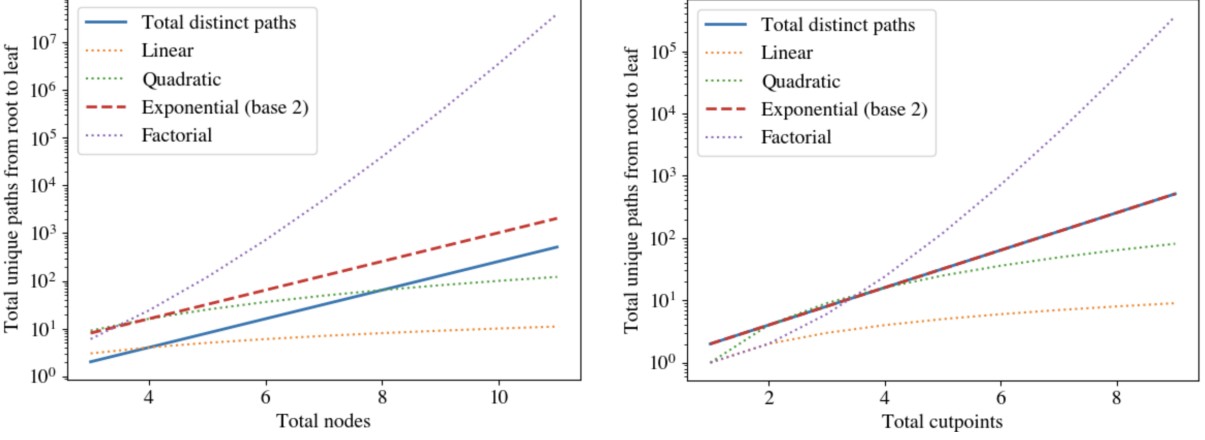

*Figure D.3.* Total unique paths from root to leaf in a CCT with respect to total nodes (left) and total cutpoints (right).

# E. SIMULATIONS

## E.1. Numerical Behavior in Finite Samples

The following experiments demonstrate the numerical behavior of inductive and deductive CCR for (1) the PNS when all cause-effect pairs satisfy the monotonicty assumption and (2) the ATE in linear SCMs. All simulations were performed on a MacBook Pro (Apple M2 Pro).

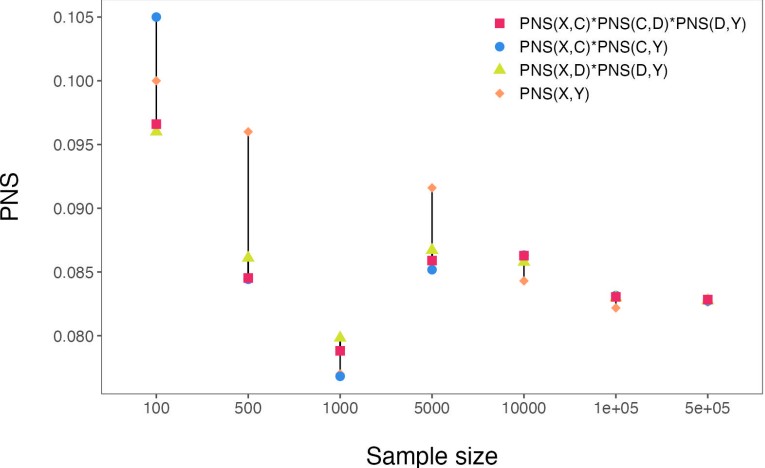

*Figure E.1.* Finite sample simulations of internally consistent inductive CCR for the PNS. Causal functions were logical *or*. For the CCT in Figure 4, all compositions converged to the same value as sample size increased. See Figure E.3 for analogous results for the ATE in linear-Gaussian SCMs.

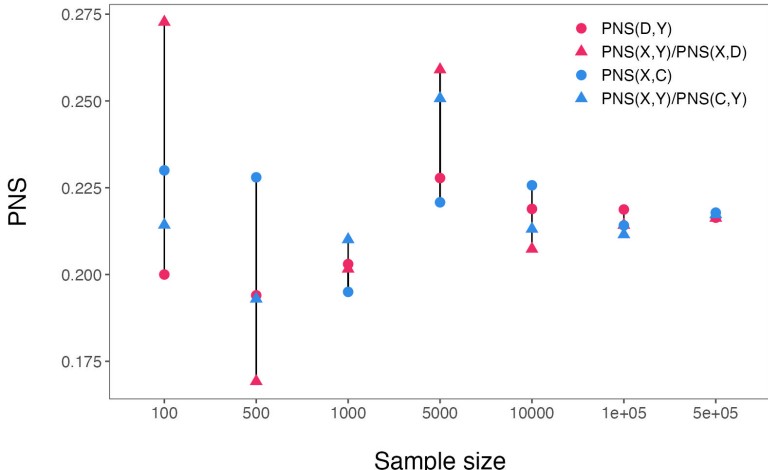

*Figure E.2.* Finite sample simulations of internally consistent deductive CCR for the PNS. Causal functions were logical *and*. For the CCT in Figure 4, points of the same color were expected to converge. See Figure E.4 for analogous results for the ATE in linear-Gaussian SCMs.

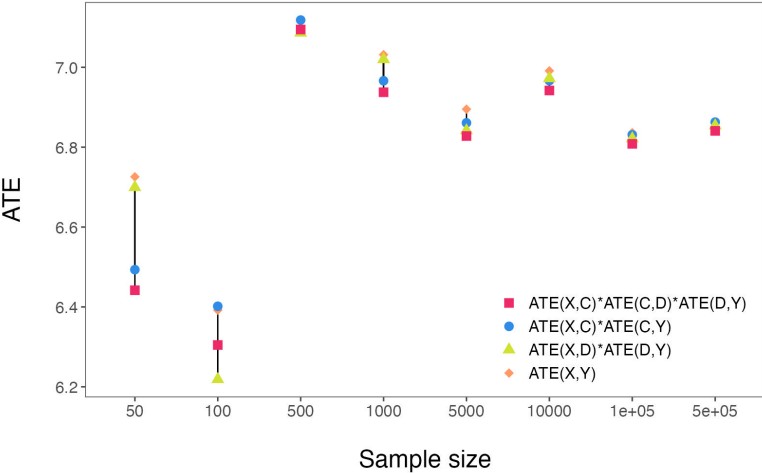

*Figure E.3.* Finite sample simulations of internally consistent inductive CCR for the ATE in linear-Gaussian SCMs. For the CCT in Figure 4, compositions converged as sample size increased. Estimates were obtained by linear regression (https://scikit-learn.org/).

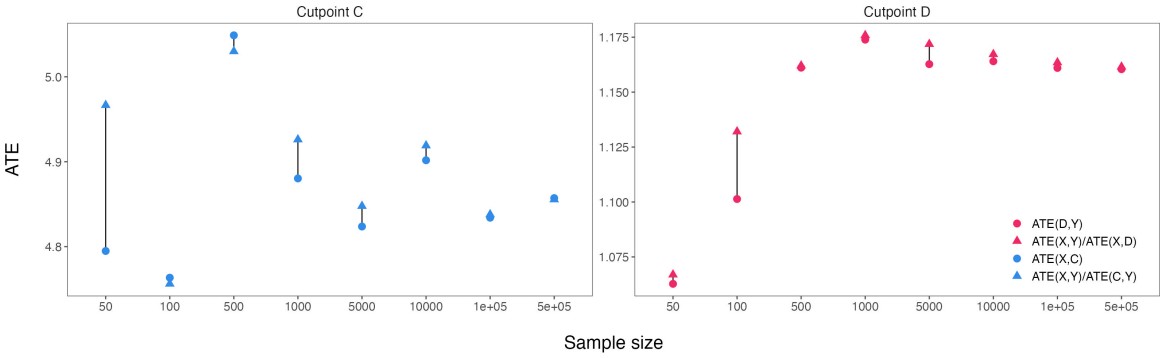

*Figure E.4.* Finite sample simulations of internally consistent deductive CCR for the ATE in linear-Gaussian SCMs. For the CCT in Figure 4, compositions converged as sample size increased. Estimates were obtained by linear regression (https://scikit-learn.org/).

# F. AUTOMATED TASK GENERATOR

To facilitate future benchmarking, we release open-source code for automated CCR task generation (https://jmaasch.github.io/ccr/). Tasks are of the same form as the running example used in this work. Code provides the following functionality:

- Generate causal DAG.
- Generate its corresponding CCT.
- Enumerate quantities of interest for CCR evaluation: global, local, and compositions.
- Generate factual and counterfactual text prompts corresponding to the generated SCM and chosen theme. Variables are assigned random human names. Themes are `CandyParty` (which requires quantitative reasoning; Figure F.1) and `FlowerGarden` (which does not require quantitative reasoning; Figure F.2).
- Sample observational and interventional data from the SCM.
- Compute the PNS, PN, PS, and ATE from interventional data samples.

The user can control the following sources of variation in the SCM:

- Total biconnected components in the causal DAG.
- Nodes per biconnected component.
- Graph type of biconnected component (cycle or wheel graph), which impacts connectivity.

The following elements are assigned at random:

- Parameters to Bernoulli distributions (chosen uniformly at random).
- Causal functions (a mix of logical *or* and logical *and*, chosen uniformly at random).

Each task is constructed as follows. See Figures F.1 and F.2 for concrete examples.

1. **Causal world model.** First, we define a fictional world corresponding to a randomly generated causal graph. This will be the causal world model for the LM to reason over. The structural causal model defining our fictitious world is comprised of binary exogenous noise variables, binary endogenous variables, and nonlinear causal functions (monotonic logical operators *and*, *or*).
2. **Causal context prompt.** Second, we construct a verbal description of the world model. This verbal description — our "causal context prompt" — contains all pertinent details needed for the LM to infer the world model, as well as extraneous details not needed to solve the CCR task. The causal context centers on a user defined theme (e.g., `CandyParty`, `FlowerGarden`, etc.).
3. **Sampling**. Third, we randomly sample exogenous variables and extraneous variables and compute true endogenous variable values. Sampled values are then used to construct the "sample context" in natural language, which is concatenated to our causal context prompt. Each causal context will be copied many times, where each copy is paired with a new sample context.
4. **Factual query prompts.** Next, we construct factual queries by treating the causal context + sample context as observational data. All queries are phrased as yes/no questions. The factual query is then concatenated to a copy of the causal context + sample context.
5. **Counterfactual query prompts.** Finally, we pair each factual prompt with an interventional query corresponding to the appropriate counterfactual ($do(X = \text{TRUE})$ or $do(X = \text{FALSE})$). The resulting counterfactual query is then concatenated to a copy of the causal context + sample context. As with factual queries, all counterfactual queries are phrased as yes/no questions.

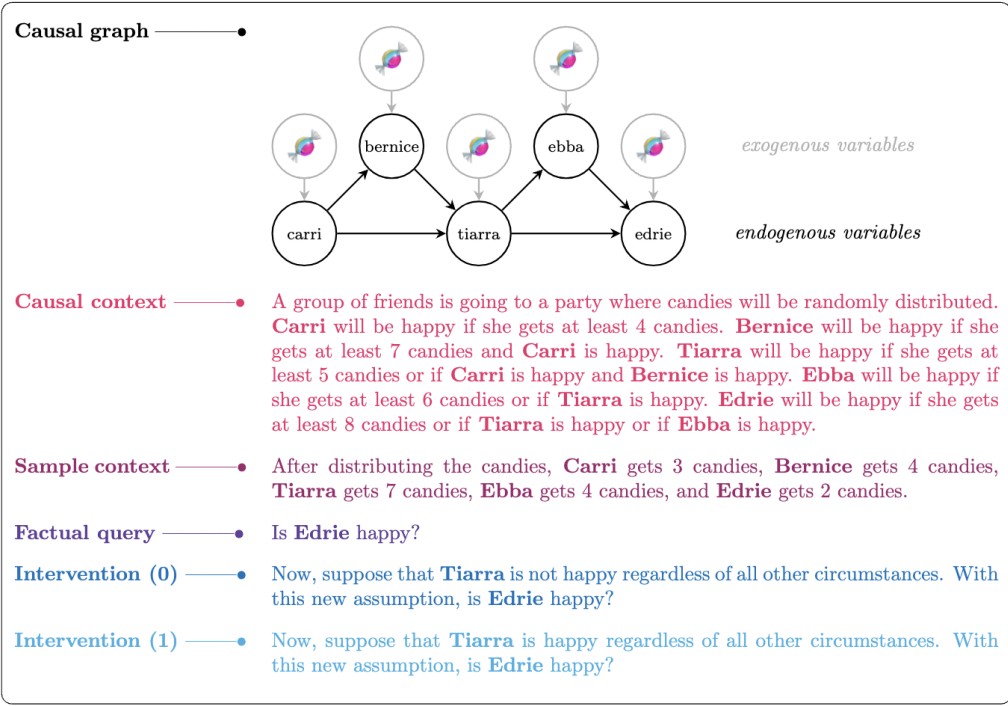

*Figure F.1.* A simple `CandyParty` prompt, which requires quantitative reasoning.

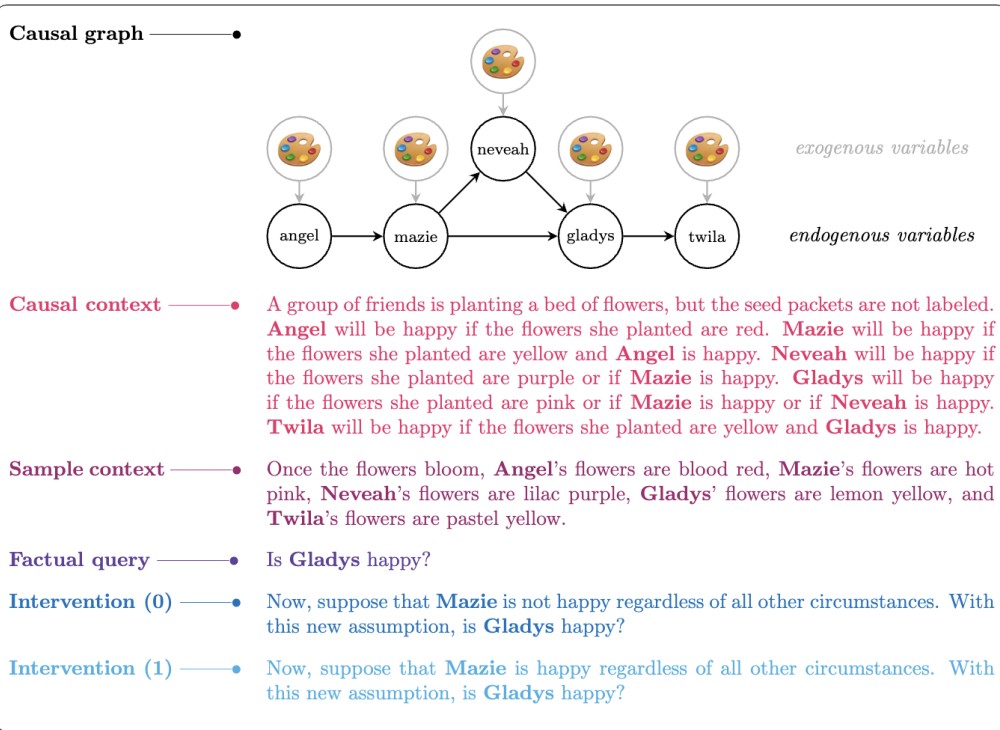

*Figure F.2.* A simple `FlowerGarden` prompt, which requires qualitative reasoning.

# G. LM EXPERIMENTS

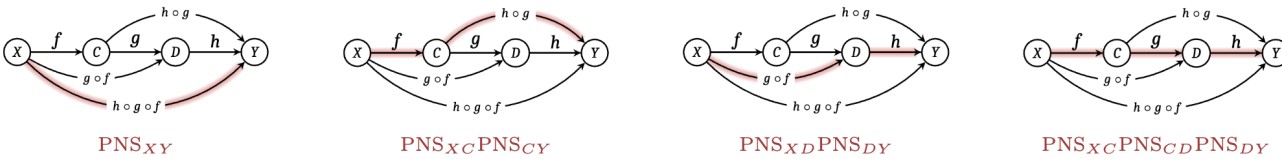

*Figure G.1.* All paths from $X$ to $Y$ in the CCT corresponding to the DAG in Figure 4, where $f \coloneqq \text{PNS}_{XC}$, $g \coloneqq \text{PNS}_{CD}$, $h \coloneqq \text{PNS}_{DY}$, $g \circ f \coloneqq \text{PNS}_{XD}$, $h \circ g \circ f \coloneqq \text{PNS}_{XY}$, etc., and composition is multiplicative.

| MODEL | PARAMETERS | LINK |
|---|---|---|
| Phi-3-Mini-128K-Instruct (Abdin et al., 2024) | 3.82B | https://huggingface.co/microsoft/Phi-3-mini-128k-instruct |
| Llama-2-7b-Chat-HF (Touvron et al., 2023) | 6.74B | https://huggingface.co/meta-llama/Llama-2-7b-chat-hf |
| Llama-3-8B-Instruct (Dubey et al., 2024) | 8.03B | https://huggingface.co/meta-llama/Meta-Llama-3-8B-Instruct |
| Llama-3.1-8B-Instruct (Dubey et al., 2024) | 8.03B | https://huggingface.co/meta-llama/Llama-3.1-8B-Instruct |
| OpenMath2-Llama3.1-8B (Toshniwal et al., 2024) | 8.03B | https://huggingface.co/nvidia/OpenMath2-Llama3.1-8B |
| GPT-4o | > 175B | https://openai.com/index/gpt-4o-system-card/ |
| o1 | > 175B | https://openai.com/o1/ |

*Table G.1.* Large language models used for inference. The exact number of parameters in GPT-4o and o1 is not public knowledge, so we note the size of GPT-3 as a lower bound (B denotes billions).

## G.1. Experimental Design

**Model Inference** We used a single A100 GPU for all experiments. Models used for inference were LMs fine-tuned for dialogue, as reported in Table G.1. Llama 3.1 Math was also fine-tuned for math reasoning. OpenAI's o1 model is marketed as a "high intelligence reasoning model."[4] We did not perform any additional fine-tuning. All models used default hyperparameters for response generation except when greedy decoding was the default generation strategy, in which case default Hugging Face hyperparameters were used for sampling. For Boolean extraction, greedy decoding was used.

**Translating Causal Queries to Text** We translated the data generating process represented by the DAG in Figure 4 to a mathematical word problem, as expressed in the prompt in Figure G.2. This prompt is based on the `CandyParty` problem described in González & Nori (2024) and Hüyük et al. (2025). Additionally, we implement a CoT wrapper for our original prompt template that uses two examples to demonstrate CoT for the model: one factual and one counterfactual (Figure G.3). Questions and answers used in the CoT scenario were sampled and calculated identically to the non-CoT experiments.

We define SCM $\mathcal{M} \coloneqq \langle \mathbf{V}, \mathbf{U}, \mathcal{F}, p(\mathbf{u}) \rangle$, where $\mathbf{V}$ are binary variables representing the nodes in Figure 4, causal functions $f \in \mathcal{F}$ are logical *or* ($\vee$), and distribution $p(\mathbf{u})$ is Bernoulli. Logical *or* is a monotone Boolean function, satisfying the monotonicity condition for point identifiability of the PNS from causal effects (Definition 2.5).

Each node of the ground truth graph is treated as a person in our word problem: $X$ = Xinyu, $A$ = Ara, $B$ = Becca, $C$ = Celine, $D$ = Daphne, $E$ = Emma, $F$ = Fox, $Y$ = Yasmin. Values for all $V_i \in \mathbf{V}$ are given by

$$v_i = pa_1, \vee ... \vee pa_k \vee \text{Ber}(0.1\, T_{V_i}) \tag{G.1}$$

where $\{pa_j\}_{j=1}^{k}$ are the $k$ parents of $V_i$. All $\{T \cdot\}$ in the context prompt take a value of 7, such that all exogenous variables are drawn from Bernoulli distributions parameterized by $p = 0.7$.

Examples of factual and counterfactual questions and responses are given in Figures G.4 and G.5 (see also: Figures F.1, F.2). For counterfactual questions, a new assumption was introduced about the variable acting as the cause. The model was then asked a question about the variable acting as the effect.

---

[4] https://platform.openai.com/docs/guides/reasoning

**Extracting Boolean Values from Text**    To compute the PNS, model text responses were first translated to the corresponding Boolean answer. The corresponding Boolean was extracted using Llama 3 with greedy decoding, given the following prompt: "I will give you a question and its answer. Determine whether the meaning of the answer is 'TRUE' or 'FALSE'. An answer is 'TRUE' if it contains phrases like 'yes', 'it holds', 'correct', 'true', or similar affirmations. An answer is 'FALSE' if it contains phrases like 'no', 'it does not hold', 'incorrect', 'false', or similar negations. Respond only with one word: 'TRUE' or 'FALSE'. Question: 'q' Answer: 'a' Is the meaning 'TRUE' or 'FALSE'?"

**Computing PNS Estimates**    For each cause-effect pair, we sampled 1000 sets of exogenous variable values. For each exogenous value set, we generated one factual and one counterfactual question about the effect of interest using the prompt template as context (Figure G.2). Five replicate LM responses were then sampled for each question. From these 10,000 total responses (5000 factual and 5000 counterfactual), 1000 factual responses and 1000 counterfactual responses were randomly subsampled (one per set of five replicate responses). The subsample of factual and counterfactual responses was then used to compute the PNS. This subsampling procedure was repeated 1000 times, resulting in a distribution of 1000 PNS estimates per cause-effect pair per model.

**Approximation Errors**    Approximation errors were computed as the relative absolute error (RAE) for each individual PNS estimate. The RAE is the absolute error (AE) normalized by the true PNS. When comparing errors across different quantities (e.g., $PNS_{XY}$ versus $PNS_{XC}$), normalization is needed. Thus, we generally only report the RAE. For external validity, these metrics are computed with respect to ground truth ($PNS^*\cdot$).

$$AE_{\text{external}} := \mid PNS^*\cdot - \widehat{PNS\cdot} \mid \tag{G.2}$$

$$RAE_{\text{external}} := \frac{\mid PNS^*\cdot - \widehat{PNS\cdot} \mid}{PNS^*\cdot} \tag{G.3}$$

For internal consistency, these metrics are computed using estimates for two equivalent quantities ($\widehat{PNS\cdot}$ and $\widehat{PNS\cdot}'$).

$$AE_{\text{internal}} := \mid \widehat{PNS\cdot} - \widehat{PNS\cdot}' \mid \tag{G.4}$$

$$RAE_{\text{internal}} := \frac{\mid \widehat{PNS\cdot} - \widehat{PNS\cdot}' \mid}{\widehat{PNS\cdot}} \tag{G.5}$$

---

*Prompt*

```
Xinyu, Ara, Becca, Celine, Daphne, Emma, Fox, and Yasmin are going to a party,
where the host is going to distribute candies.  Xinyu will be happy if she gets at
least {TX} candies.  Ara will be happy if Xinyu is happy or if he gets at least {TA}
candies.  Becca will be happy if Xinyu is happy or if she gets at least {TB} candies.
Celine will be happy if Xinyu is happy or if Ara is happy or if Becca is happy or if
she gets at least {TC} candies.  Daphne will be happy if Celine is happy or if she
gets at least {TD} candies.  Emma will be happy if Daphne is happy or if she gets at
least {TE} candies.  Fox will be happy if Daphne is happy or if Emma is happy or if
he gets at least {TF} candies.  Yasmin will be happy if Emma is happy or if Fox is
happy or if she gets at least {TY} candies.  After distributing the candies, Xinyu
gets {NX}, Ara gets {NA}, Becca gets {NB}, Celine gets {NC}, Daphne gets {ND}, Emma
gets {NE}, Fox gets {NF}, and Yasmin gets {NY}.
```

*Figure G.2.* Context prompt template for inductive CCR evaluation. For all experiments reported in this work, {T·} = 7.

---

*Prompt*

**QUESTION:** Xinyu, Ara, Becca, Celine, Daphne, Emma, Fox, and Yasmin are going to a party, where the host is going to distribute candies. Xinyu will be happy if she gets at least 3 candies. Ara will be happy if Xinyu is happy or if he gets at least 4 candies. Becca will be happy if Xinyu is happy or if she gets at least 10 candies. Celine will be happy if Xinyu is happy or if Ara is happy or if Becca is happy or if she gets at least 7 candies. Daphne will be happy if Celine is happy or if she gets at least 5 candies. Emma will be happy if Daphne is happy or if she gets at least 6 candies. Fox will be happy if Daphne is happy or if Emma is happy or if he gets at least 10 candies. Yasmin will be happy if Emma is happy or if Fox is happy or if she gets at least 1 candies. After distributing the candies, Xinyu gets 6, Ara gets 7, Becca gets 5, Celine gets 1, Daphne gets 3, Emma gets 4, Fox gets 9, and Yasmin gets 8. Is Becca happy? Be as concise as possible.
**ANSWER:** Since Xinyu gets 6 candies, which is more than 3, she is happy. In that case, Becca is happy no matter how many candies she got, because she will be happy if Xinyu is happy. Therefore, yes, Becca is happy!
**QUESTION:** Xinyu, Ara, Becca, Celine, Daphne, Emma, Fox, and Yasmin are going to a party, where the host is going to distribute candies. Xinyu will be happy if she gets at least 3 candies. Ara will be happy if Xinyu is happy or if he gets at least 4 candies. Becca will be happy if Xinyu is happy or if she gets at least 10 candies. Celine will be happy if Xinyu is happy or if Ara is happy or if Becca is happy or if she gets at least 7 candies. Daphne will be happy if Celine is happy or if she gets at least 5 candies. Emma will be happy if Daphne is happy or if she gets at least 6 candies. Fox will be happy if Daphne is happy or if Emma is happy or if he gets at least 10 candies. Yasmin will be happy if Emma is happy or if Fox is happy or if she gets at least 1 candies. After distributing the candies, Xinyu gets 6, Ara gets 7, Becca gets 5, Celine gets 1, Daphne gets 3, Emma gets 4, Fox gets 9, and Yasmin gets 8. Now, suppose that Xinyu is not happy regardless of the candy distribution. Is Becca happy? Be as concise as possible.
**ANSWER:** Since Xinyu gets 6 candies, which is more than 3, she is happy. However, we are asked to assume Xinyu is not happy regardless. In that case, Becca is happy only if she gets at least 10 candies. Becca gets 5 candies, which is less than 10. Therefore, no, Becca is not happy!
**QUESTION:** Xinyu, Ara, Becca, Celine, Daphne, Emma, Fox, and Yasmin are going to a party, where the host is going to distribute candies. Xinyu will be happy if she gets at least 7 candies. Ara will be happy if Xinyu is happy or if he gets at least 7 candies. Becca will be happy if Xinyu is happy or if she gets at least 7 candies. Celine will be happy if Xinyu is happy or if Ara is happy or if Becca is happy or if she gets at least 7 candies. Daphne will be happy if Celine is happy or if she gets at least 7 candies. Emma will be happy if Daphne is happy or if she gets at least 7 candies. Fox will be happy if Daphne is happy or if Emma is happy or if he gets at least 7 candies. Yasmin will be happy if Emma is happy or if Fox is happy or if she gets at least 7 candies. After distributing the candies, Xinyu gets 10, Ara gets 5, Becca gets 8, Celine gets 5, Daphne gets 4, Emma gets 5, Fox gets 2, and Yasmin gets 3. Now, suppose that Xinyu is not happy regardless of the candy distribution. With this assumption, is Yasmin happy? Be as concise as possible.
**ANSWER:**

*Figure G.3.* Example CoT formulation of the original `CandyParty` prompt, where two examples that demonstrate CoT are provided to the model: one factual and one counterfactual.

*Prompt*

After distributing the candies, Xinyu gets 4, Ara gets 6, Becca gets 5, Celine gets 10, Daphne gets 1, Emma gets 1, Fox gets 4, and Yasmin gets 3. Is Celine happy? Be as concise as possible.

*Response*

No, Celine is not happy. She wants at least 7 candies but got only 10.

*Figure G.4.* Factual question and answer from Llama 2, displaying incorrect numerical reasoning.

*Prompt*

Now, suppose that Xinyu is happy regardless of the candy distribution. With this assumption, is Celine happy? Be as concise as possible.

*Response*

Under the assumption that Xinyu is happy regardless of the candy distribution, Celine is not happy.

Reason: Since Xinyu is happy regardless of the candy distribution, it means that she got at least 7 candies. However, Celine only got 10 candies, which is less than 7. Therefore, Celine is not happy.

*Figure G.5.* Counterfactual question and answer from Llama 2, displaying incorrect numerical reasoning.

*Prompt*

After distributing the candies, Xinyu gets 10, Ara gets 6, Becca gets 7, Celine gets 6, Daphne gets 1, Emma gets 10, Fox gets 10, and Yasmin gets 2. Is Daphne happy? Be as concise as possible.

*Response*

Daphne will be happy if Celine is happy or if she gets at least 7 candies. Since Daphne gets less than 7 candies (1), she won't be happy based on the candies she gets. Celine gets 6 candies, which is less than 7. Since neither of Daphne's conditions is met, Daphne is not happy.

*Figure G.6.* A case where Llama 3.1 failed to follow the full chain of logic. The correct line of logic is as follows: *Daphne will be happy if Celine is happy or if she gets at least 7 candies. Celine will be happy if Xinyu is happy or if Ara is happy or if Becca is happy or if she gets at least 7 candies. Xinyu will be happy if she gets at least 7 candies. Since Xinyu received 10 candies, she is happy, and so should be Celine. Thus, Daphne should also be happy.*

---

*Prompt*

Xinyu, Ara, Becca, Celine, Daphne, Emma, Fox, and Yasmin are going to a party, where the host is going to distribute candies. Xinyu will be happy if she gets at least 7 candies. Ara will be happy if Xinyu is happy or if he gets at least 7 candies. Becca will be happy if Xinyu is happy or if she gets at least 7 candies. Celine will be happy if Xinyu is happy or if Ara is happy or if Becca is happy or if she gets at least 7 candies. Daphne will be happy if Celine is happy or if she gets at least 7 candies. Emma will be happy if Daphne is happy or if she gets at least 7 candies. Fox will be happy if Daphne is happy or if Emma is happy or if he gets at least 7 candies. Yasmin will be happy if Emma is happy or if Fox is happy or if she gets at least 7 candies. After distributing the candies, Xinyu gets 6, Ara gets 9, Becca gets 7, Celine gets 7, Daphne gets 5, Emma gets 5, Fox gets 4, and Yasmin gets 1. Is Yasmin happy? Be as concise as possible.

---

*Response*

Xinyu is not happy because she gets 6 candies, which is less than 7. Ara is happy because he gets 9 candies, which is more than 7. Becca is happy because she gets 7 candies, which is exactly 7. Celine is happy because Ara is happy. **Daphne is not happy because she gets 5 candies, which is less than 7, and Celine is happy.** Emma is not happy because she gets 5 candies, which is less than 7, and Daphne is not happy. Fox is not happy because he gets 4 candies, which is less than 7, and neither Daphne nor Emma is happy. Yasmin is not happy because she gets 1 candy, which is less than 7, and neither Emma nor Fox is happy. Therefore, no, Yasmin is not happy!

---

*Figure G.7.* A case where GPT-4o with CoT failed to follow the full chain of logic given a factual prompt used to compute $\widehat{\text{PNS}}_{XY}$. The first reasoning error is in bold. In reality, Daphne is happy if her causal parent Celine is happy or if she receives at least 7 candies. The model correctly reasons that the number of candies given to Daphne does not satisfy these conditions (5 is less than 7). Despite the fact that the model acknowledges that Celine is happy twice, it contradicts itself and incorrectly reasons that Daphne is unhappy. This error then propagates through the remaining logic.

---

*Prompt*

Xinyu, Ara, Becca, Celine, Daphne, Emma, Fox, and Yasmin are going to a party, where the host is going to distribute candies. Xinyu will be happy if she gets at least 7 candies. Ara will be happy if Xinyu is happy or if he gets at least 7 candies. Becca will be happy if Xinyu is happy or if she gets at least 7 candies. Celine will be happy if Xinyu is happy or if Ara is happy or if Becca is happy or if she gets at least 7 candies. Daphne will be happy if Celine is happy or if she gets at least 7 candies. Emma will be happy if Daphne is happy or if she gets at least 7 candies. Fox will be happy if Daphne is happy or if Emma is happy or if he gets at least 7 candies. Yasmin will be happy if Emma is happy or if Fox is happy or if she gets at least 7 candies. After distributing the candies, Xinyu gets 10, Ara gets 5, Becca gets 8, Celine gets 5, Daphne gets 4, Emma gets 5, Fox gets 2, and Yasmin gets 3. Now, suppose that Xinyu is not happy regardless of the candy distribution. With this assumption, is Yasmin happy? Be as concise as possible.

---

*Response*

If Xinyu is not happy regardless of the candy distribution, then she is not happy even though she gets 10 candies. In this case, Yasmin will be happy if Emma is happy, Fox is happy, or if she gets at least 7 candies. Yasmin gets 3 candies, which is less than 7, so she is not happy based on her own candies.

Now, let's check if Emma or Fox is happy. Emma will be happy if Daphne is happy or if she gets at least 7 candies. Emma gets 5 candies, which is less than 7, and **Daphne is not happy because she gets 4 candies, which is less than 7, and Celine is not happy either**. Therefore, Emma is not happy.

Fox will be happy if Daphne is happy, Emma is happy, or if he gets at least 7 candies. Fox gets 2 candies, which is less than 7, and since neither Daphne nor Emma is happy, Fox is not happy.

Since neither Emma nor Fox is happy, Yasmin is not happy. Therefore, no, Yasmin is not happy.

---

*Figure G.8.* A case where GPT-4o with CoT failed to follow the full chain of logic given a counterfactual prompt used to compute $\widehat{\text{PNS}}_{XY}$. An intervention is placed on Xinyu such that she is unhappy despite receiving enough candies to be happy. The first reasoning error is in bold. While the model correctly extracts the causal parents of Yasmin, Emma, and Fox, it does not express understanding of the causal structure upstream. In reality, Becca is happy because she received greater than 7 candies, which is sufficient to make Celine happy. In turn, this is sufficient to make Daphne happy. However, the model reasons that both Daphne and Celine are unhappy, without providing further explanation. This error then propagates through the remaining logic.

---

*Prompt*

Xinyu, Ara, Becca, Celine, Daphne, Emma, Fox, and Yasmin are going to a party, where
the host is going to distribute candies.  Xinyu will be happy if she gets at least
7 candies.  Ara will be happy if Xinyu is happy or if he gets at least 7 candies.
Becca will be happy if Xinyu is happy or if she gets at least 7 candies.  Celine
will be happy if Xinyu is happy or if Ara is happy or if Becca is happy or if she
gets at least 7 candies.  Daphne will be happy if Celine is happy or if she gets at
least 7 candies.  Emma will be happy if Daphne is happy or if she gets at least 7
candies.  Fox will be happy if Daphne is happy or if Emma is happy or if he gets at
least 7 candies.  Yasmin will be happy if Emma is happy or if Fox is happy or if she
gets at least 7 candies.  After distributing the candies, Xinyu gets 6, Ara gets 8,
Becca gets 6, Celine gets 7, Daphne gets 2, Emma gets 4, Fox gets 3, and Yasmin gets
2.  Is Yasmin happy?  Be as concise as possible.

---

*Response*

Xinyu is not happy because she gets 6 candies, which is less than 7.  Ara is happy
because he gets 8 candies, which is more than 7.  Celine is happy because Ara is
happy.  **Daphne is not happy because she gets 2 candies, which is less than 7, and
Celine is the only one who can make her happy.**  Emma is not happy because she gets 4
candies, which is less than 7, and Daphne is not happy.  Fox is not happy because he
gets 3 candies, which is less than 7, and neither Daphne nor Emma is happy.  Yasmin
is not happy because she gets 2 candies, which is less than 7, and neither Emma nor
Fox is happy.  Therefore, Yasmin is not happy.

---

*Figure G.9.* A case where GPT-4o with CoT failed to reason correctly despite extracting the correct causal relations from a factual prompt used to compute $\widehat{\mathrm{PNS}}_{DY}$. The first reasoning error is in bold. As in Figure G.7, the model correctly implies that Celine (who is happy) is the only causal parent of Daphne in the underlying DAG, and yet still incorrectly concludes that Daphne is unhappy. This error then propagates through the remaining logic. This same error was observed multiple times from GPT-4o with CoT.

---

*Prompt*

Xinyu, Ara, Becca, Celine, Daphne, Emma, Fox, and Yasmin are going to a party, where
the host is going to distribute candies.  Xinyu will be happy if she gets at least
7 candies.  Ara will be happy if Xinyu is happy or if he gets at least 7 candies.
Becca will be happy if Xinyu is happy or if she gets at least 7 candies.  Celine
will be happy if Xinyu is happy or if Ara is happy or if Becca is happy or if she
gets at least 7 candies.  Daphne will be happy if Celine is happy or if she gets
at least 7 candies.  Emma will be happy if Daphne is happy or if she gets at least
7 candies.  Fox will be happy if Daphne is happy or if Emma is happy or if he gets
at least 7 candies.  Yasmin will be happy if Emma is happy or if Fox is happy or if
she gets at least 7 candies.  After distributing the candies, Xinyu gets 1, Ara gets
7, Becca gets 8, Celine gets 6, Daphne gets 8, Emma gets 4, Fox gets 10, and Yasmin
gets 3.  Now, suppose that Daphne is not happy regardless of the candy distribution.
With this assumption, is Yasmin happy?  Be as concise as possible.

---

*Response*

Since Daphne is not happy regardless, **Yasmin can only be happy if Emma is happy
or if she gets at least 7 candies.**  Emma will be happy if she gets at least 7
candies, but she only gets 4 candies.  Yasmin gets 3 candies, which is less than 7.
Therefore, no, Yasmin is not happy!

---

*Figure G.10.* A case where GPT-4o with CoT failed to extract causal relations correctly when given a counterfactual prompt used to compute $\widehat{\mathrm{PNS}}_{DY}$. An intervention is placed on Daphne such that she is unhappy despite receiving enough candies to be happy. The first reasoning error is in bold. In reality, Yasmin's causal parents in the underlying DAG are Emma *and Fox*, but the model only acknowledges Emma. Though the model correctly reasons that Emma is unhappy, Fox is happy despite Daphne's unhappiness because he has received $10 (\geq 7)$ candies. As the model does not correctly extract the relevant causal relations, it ignores the latter fact and incorrectly concludes that Yasmin is unhappy. This same error persisted across multiple replicates.

## G.2. Results

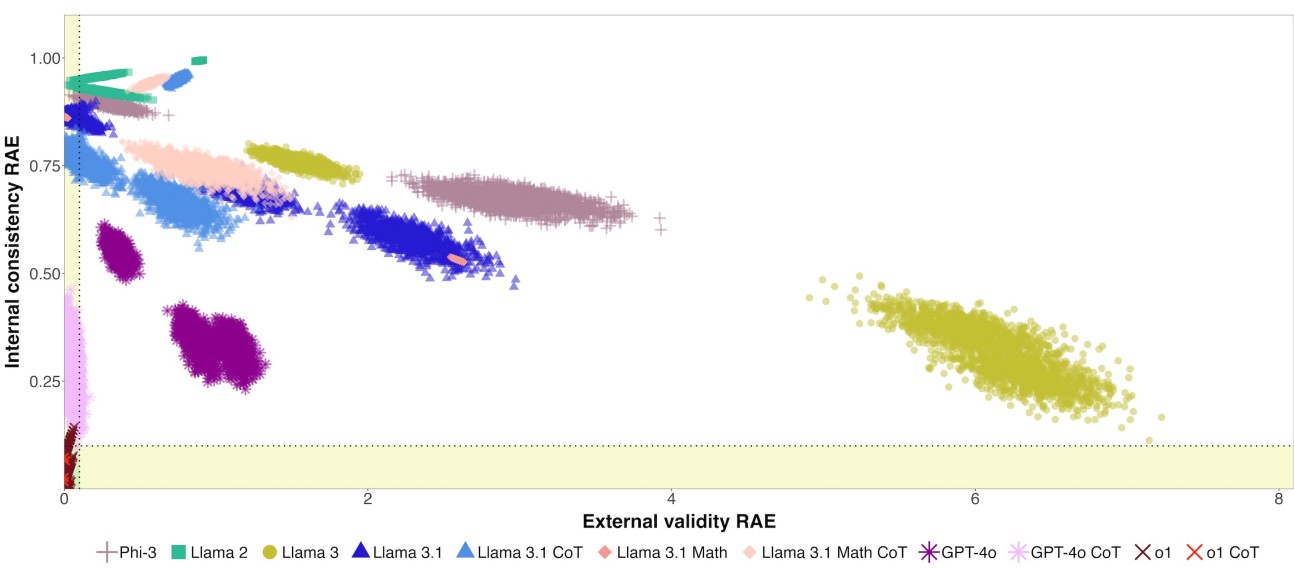

*Figure G.11.* [Full distribution for Fig. 5.] For PNS compositions ($n = 1000$ estimates per quantity per model), we compare RAE w.r.t. ground truth (external validity) and $\widehat{\text{PNS}}_{XY}$ (internal consistency) to visualize our four reasoning quadrants (VI/IC in yellow; VC in green; II in white). Dotted lines are error thresholds (RAE = 0.1). Models are listed by increasing size (Table G.1).

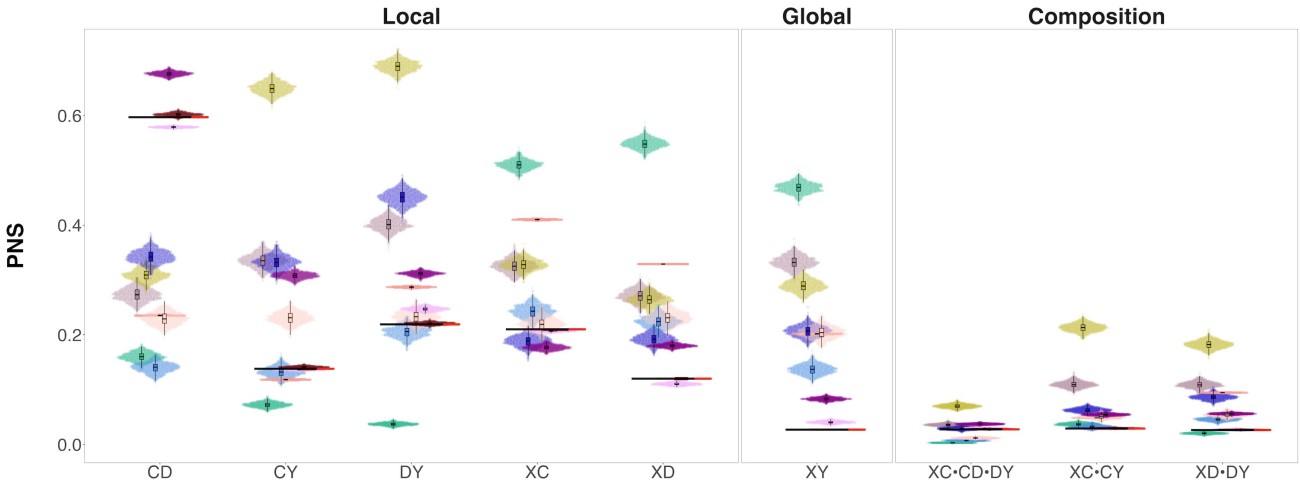

*Figure G.12.* Estimated PNS distributions ($n = 1000$) for the quantities given in Table 1, denoted here by cause-effect pair (e.g., $XC \cdot CD \cdot DY$ denotes $PNS_{XC} PNS_{CD} PNS_{DY}$). Bold black line segments represent ground truth values.

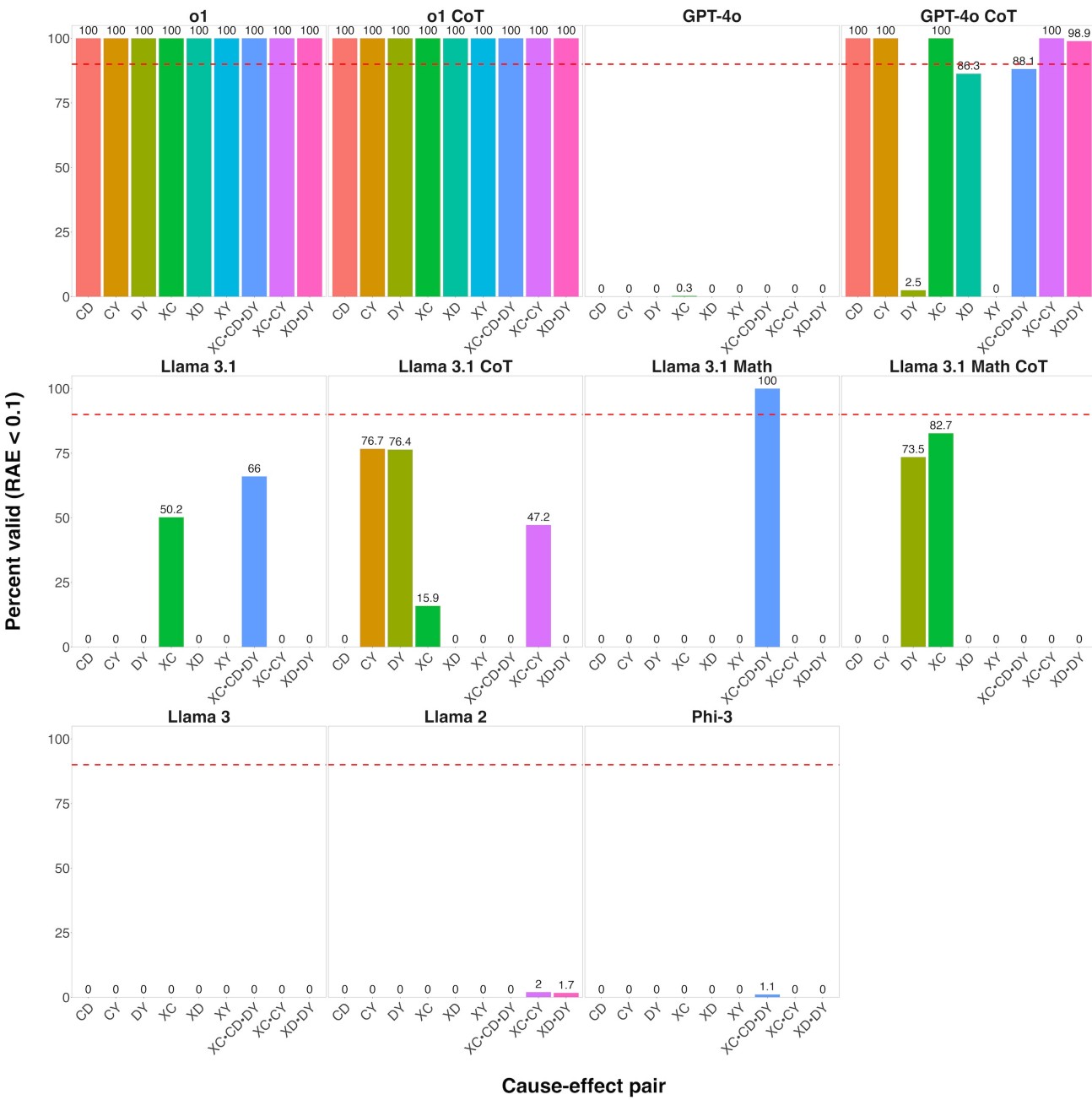

*Figure G.13.* Percent of PNS estimates ($n = 1000$) that are externally valid. Reasoning was considered externally valid for a quantity if $\geq 90\%$ of estimates had RAE $\leq 0.1$ (represented by the red dashed line). PNS are denoted by cause-effect pair (e.g., $XC \cdot CD \cdot DY$ denotes $PNS_{XC}PNS_{CD}PNS_{DY}$).

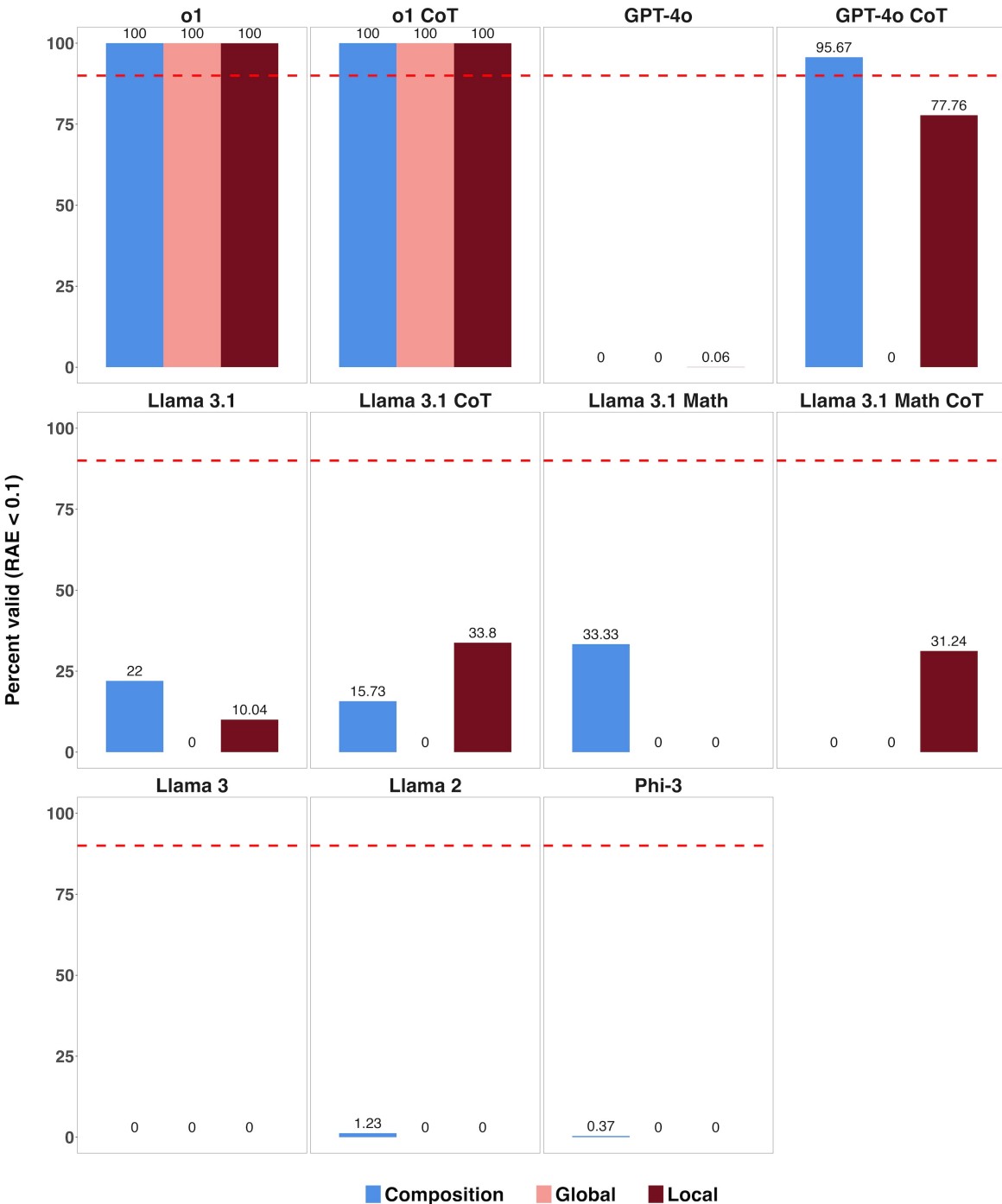

*Figure G.14.* Percent of PNS estimates ($n = 1000$) that are externally valid. Reasoning was considered externally valid for a quantity if $\geq 90\%$ of estimates had RAE $\leq 0.1$ (represented by the red dashed line).

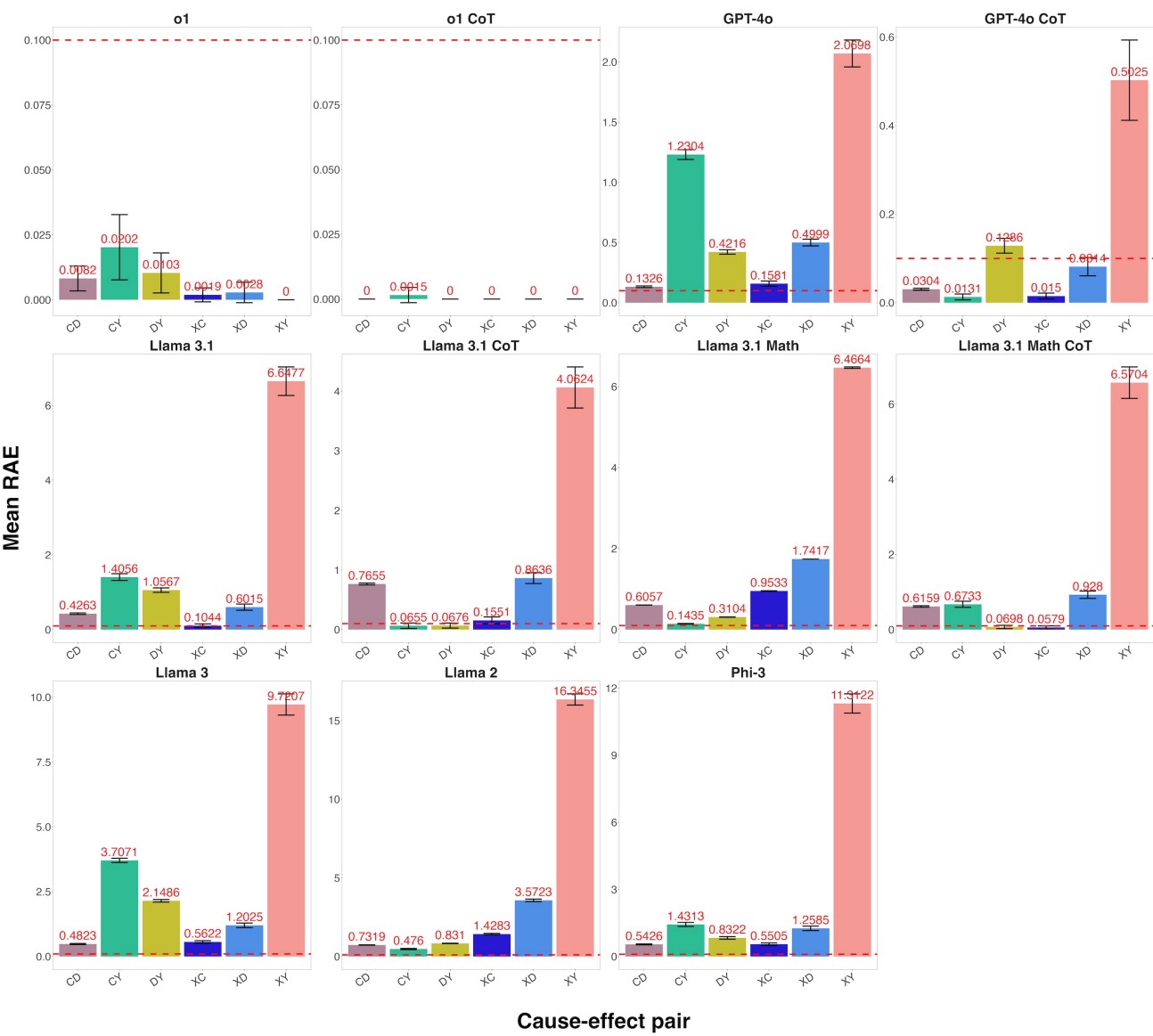

*Figure G.15.* Mean external validity RAE ($n = 1000$). Error bars represent standard deviations. An estimate was considered externally valid for a quantity if RAE $\leq 0.1$ (represented by the red dashed line).

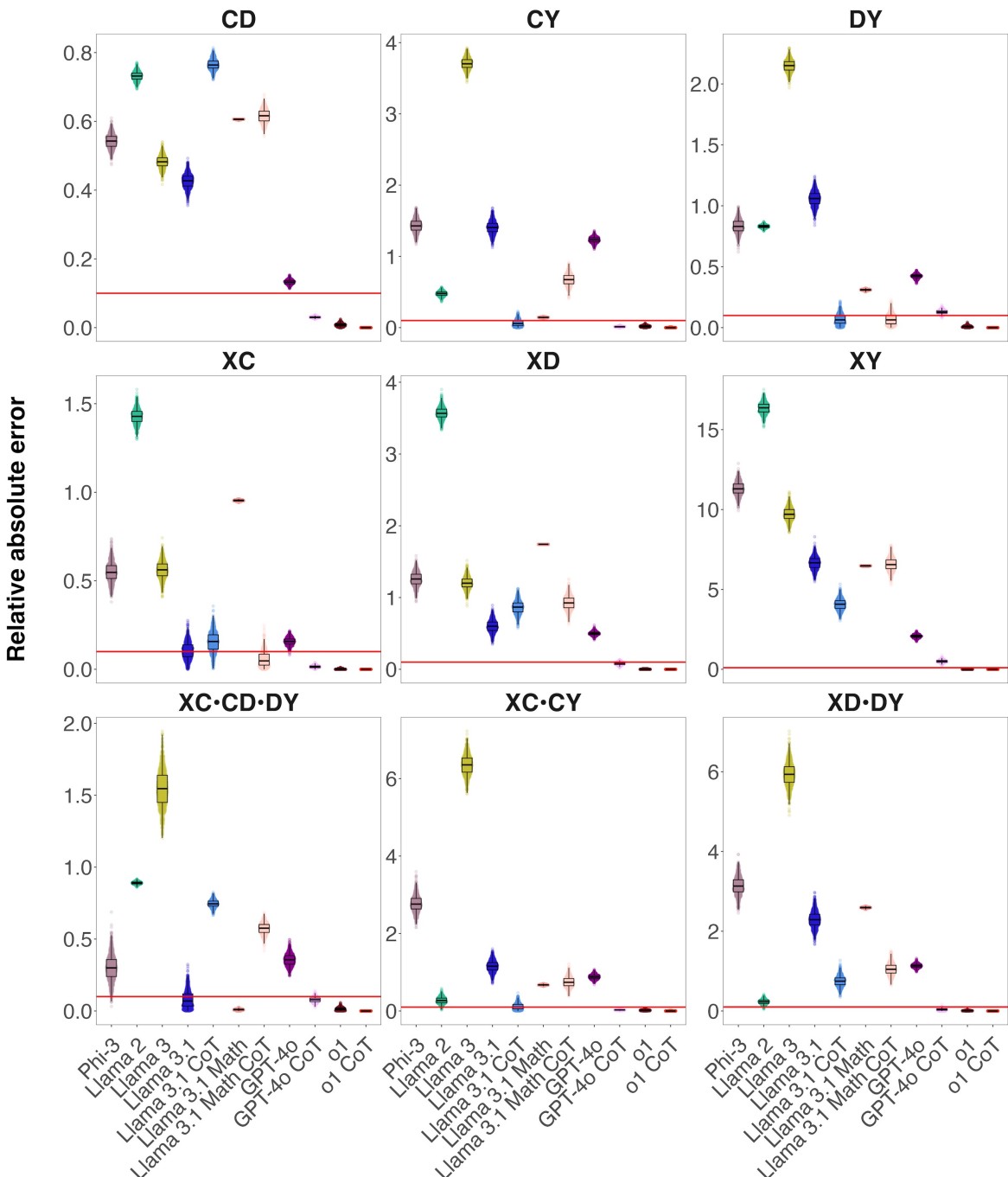

*Figure G.16.* RAE distributions for all quantities. Red lines represent the external validity cutoff (RAE = 0.1). PNS are denoted by cause-effect pair (e.g., $XC \cdot CD \cdot DY$ denotes $PNS_{XC}PNS_{CD}PNS_{DY}$).

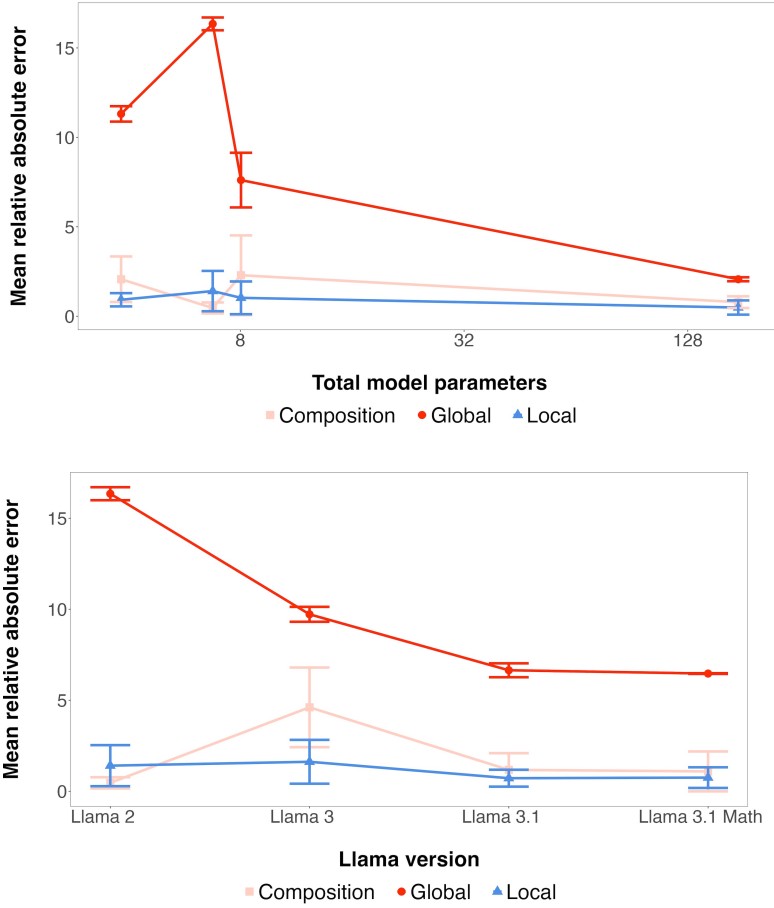

*Figure G.17.* Mean RAE (without CoT) does not consistently decrease with increasing model size ($\log_2$ scale; **left**) nor Llama version (**right**). However, errors for global quantities ($PNS_{XY}$) do monotonically decrease with increasing Llama version. Values were averaged separately for global, local, and composed quantities (Table 1). Error bars represent standard deviations. Parameter count for GPT-4o is set to the GPT-3 count (175B) due to information availability.

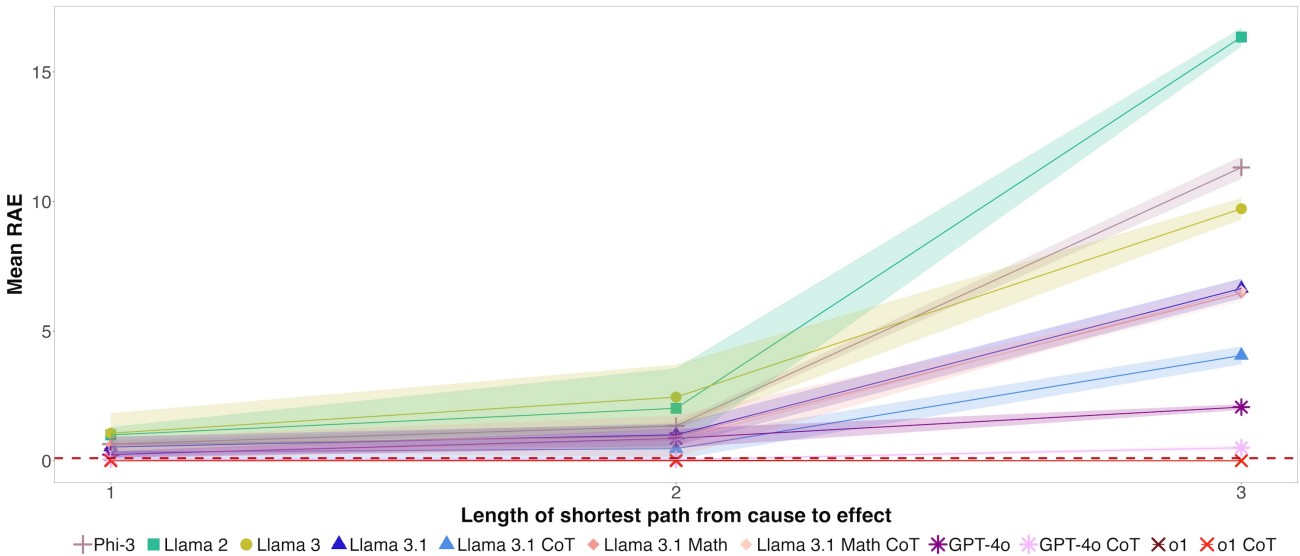

*Figure G.18.* RAE generally increases with length of shortest path from cause to effect. Red dashed line denotes external validity cutoff (RAE = 0.1), with standard deviations in shaded regions.

