# OpenReview forum: "Compositional Causal Reasoning Evaluation in Language Models"
_ICML.cc/2025/Conference — ICML 2025 poster_

### Official Review · Reviewer_GPYQ · 2025-02-20

**Overall Recommendation:** 3

**Summary:**

This paper applies the concept of compositional reasoning to causal inference, thereby introducing the notion of compositional causal reasoning (CCR). It further analyzes the relationships among different tiers of causal measures. Building on these findings, the authors propose an evaluation framework for compositional reasoning along two principal dimensions: validity and consistency. Validity represents the correctness of the answer, and consistency represents whether the model answers the global query and the local combination to get the same result. The evaluation of these two dimensions reveals the shortcomings of existing models in causal reasoning tasks.

**Claims And Evidence:**

The authors contend that more fine-grained evaluation metrics are needed to assess whether a model genuinely engages in causal reasoning. Relying solely on accuracy (i.e., the proportion of correct answers) does not suffice to determine whether reasoning processes are authentically at work.

**Essential References Not Discussed:**

None identified.

**Experimental Designs Or Analyses:**

The authors deploy a range of model sizes to estimate PNS in both factual and counterfactual scenarios. The findings suggest that task complexity (e.g., longer causal paths) generally increases the error rate. While incorporating chain-of-thought (CoT) prompts can improve the external validity of the model’s estimates, internal consistency remains an open challenge.

**Methods And Evaluation Criteria:**

The paper defines a causal reasoning task composed of multiple sub-questions and employs external validity and internal consistency as the key metrics for model evaluation. Through the “CandyParty” example, it illustrates how to estimate two causal measures—Average Treatment Effect (ATE) and Probability of Necessity and Sufficiency (PNS)—under the assumption of monotonicity and specific network structures. The authors also derive formulas that connect these metrics under those assumptions.

**Other Comments Or Suggestions:**

The evaluation results could be presented more clearly, for instance by providing a direct comparison or mapping between Figures 3 and 5 to highlight their relationship and implications.

**Other Strengths And Weaknesses:**

Strengths:

* The theoretical framework is clearly formulated.
* Experimental results show notable patterns, underscoring the importance of both validity and consistency.

Weaknesses:

* The current evaluation relies on small-scale, toy examples, thus necessitating more extensive, large-scale tests.
* The model may be overly sensitive to prompt engineering. Future work could explore advanced prompting strategies such as ReAct.

**Questions For Authors:**

Could you clarify how the “Taxonomy of Reasoners” (i.e., valid-consistent, valid-inconsistent, invalid-consistent, invalid-inconsistent) is defined? Specifically, does it refer to a single large language model’s overarching behavior, or does it represent the performance of such a model on particular sets of tasks?

**Relation To Broader Scientific Literature:**

This work introduces a potentially valuable method for evaluating whether large language models (LLMs) possess genuine causal reasoning capabilities. The proposed framework may also be beneficial for interpretability and model safety considerations.

**Theoretical Claims:**

Under the assumptions of monotonicity and particular structural constraints, the paper presents transformation formulas for ATE and PNS. These formulas are then used to construct and assess the proposed evaluation tasks.

---

> ### Author Rebuttal · Authors · 2025-03-31
>
> We thank the reviewer for acknowledging the clarity and value of our framework, as well as the importance of validity and consistency. We appreciate the suggestion that this framework has potential utility for AI interpretability and safety, areas where we see increasing crossover with causal theory. We address additional comments below.
>
> 1. **Comment: Small-scale experiments & presenting results more clearly.**
>    - **Revision 1:** To address this comment, we have extended the CandyParty experiments to include **results for o1 with and without CoT.** See revised Figures 5 and 7 at https://tinyurl.com/bdddu2vx under `/figures_and_tables/updated_figures/` and new Fig 9 (CCTs as a visualization tool for error analysis): https://tinyurl.com/yc5cf9ev
>    - **Revision 2**: As further emphasized by our new results and discussion for o1, we evaluated on a task structure that we anticipated to elicit a spectrum of CCR behavior as a means of highlighting the validity and utility of our framework (not as a benchmarking study). The updated discussion emphasizes the implications of this range of behavior, e.g.:  “Results for task T revealed three taxonomically distinct error patterns: VC (o1), near-VI (GPT-4o with CoT), and II (all remaining models). Only o1 placed mass in the VC reasoning quadrant (Fig. 5)... However, internal inconsistency made GPT-4o with CoT a near-VI reasoner: 75.1% of all estimates were near-valid, mean external validity on local quantities and compositions exceeded 77% and 95%, respectively, and yet failure to correctly infer $\mathrm{PNS}_{XY}$ revealed internally inconsistent reasoning.” See our **response to Reviewer doGN** for further discussion of VI reasoning and error analysis for GPT-4o.
>    - **Revision 3:** To facilitate future benchmarking works, we release open source **Python code for automated CCR task generation** that can be evaluated according to Algorithm 1 for the PNS, where the user can scale the graphical complexity of the causal DAG and choose from 4 themes: candy party, flu vaccination, football games, and flower gardening. The code can be viewed here, and further unit tests will be available at camera ready: https://tinyurl.com/5be5ehyp and https://tinyurl.com/4um56dab
> 2. **Comment: Sensitivity to prompt engineering.**
>    - Indeed, the potential for sensitivity to prompt engineering was the motivation for comparing CoT and non-CoT prompting in our work. Results from GPT-4o in particular demonstrate such sensitivity. While it is out of scope to consider additional prompting strategies in this work, we agree that this would be an important future direction, especially for large-scale CCR benchmarking.
> 3. **Comment: Taxonomy of reasoners.**
>    - The reviewer asks whether these reasoning categories define a “single large language model’s overarching behavior, or does it represent the performance of such a model on particular sets of tasks?” In the present work, we limit such labels to **task-dependent behavior,** though in the future more extensive benchmarking might (cautiously) enable broader claims of overarching behavior. In Section 4 we cite the prior notion of a task-metric-model triple and state that “We tether evaluation to a task-metric-model triple”, clarifying that “success on $\langle \varphi, \mathcal{M}, \mathcal{Q} \rangle$ does not imply success on $\langle \varphi, \mathcal{M}, \mathcal{Q'} \rangle$, $\langle \varphi, \mathcal{M'}, \mathcal{Q'} \rangle$, $\langle \varphi', \mathcal{M}, \mathcal{Q'} \rangle$, etc.” and “success on CCR tasks such as this is necessary but not sufficient for demonstrating that LMs can reason.”
>    - **Revision 1:** To further clarify, we have added the following to Section 6.3 Limitations & Future Directions: “This proof-of-concept limits application of our reasoning taxonomy to task-dependent observations on model behavior. However, extensive benchmarking could potentially reveal model behaviors that persist across categories of tasks. It remains to be seen if and how different LM architectures might display significant tendencies toward certain categorizations (VC, VI, IC, II).”
>    - **Revision 2:** Please see our response to Reviewer doGN for further clarification on the VI reasoning category.

---

### Official Review · Reviewer_doGN · 2025-03-09

**Overall Recommendation:** 3

**Summary:**

In their paper, the authors formalize ways to measure the ability of reasoning systems to consistently reason over compositional causal quantities. The authors propose the general task of compositional causal reasoning (CCR) which is then defined in terms of external validity (- the adherence to ground truth quantities) and internal consistency (- answer consistency over different compositions of the same quantity). The authors define metrics to qualitatively detect the presence of both quantities and create a taxonomy that feature validity and consistency (or any subset of those) to assess the ability of models to perform CCR reasoning.

The authors specifically measure reasoning compositionality in terms of combination of average treatment effect (ATE) and necessity and sufficiency (PNS). Here, the special case of multiplicative compositionality is considered and (given particular monotonicity assumptions), the authors proof alignment of ATE and PNS. To test multiplicative compositionality, commutative cut trees (CCT) are introduced which result in chains of graphical components. The correct working of the proposed metrics is evaluated on a synthetic graph.

The authors propose an algorithm to test validity and consistency under all possible combinations of compositional ATEs using the proposed CCT. Experiments are carried out over a simple synthetic graph and constructed real-world example using several Llama versions, Phi-3-mini and GPT-4o. Experiments indicate better CCR performance for more recent models, with performance degrading with the increasing number of composed components. None of the tested models seems to feature valid and consistent behavior.

**Claims And Evidence:**

The authors claim to propose novel metrics, which are able to assess the compositional causal reasoning (CCR) abilities of reasoning systems. The proposed external validity and internal consistency metrics are generally suited to measure the intended effects.

The authors leverage and proof the compositionality of ATE and PNS within a restricted setting (monotonicity, no confounding, existence of cut points, single source/leaf node). The claimed compositionality is furthermore practically validated on a synthetic graphical model.

Categorization of the tested LLMs under the proposed taxonomy seem to indicate that no model yields valid and consistent CCR behavior. Here, strong outliers within model individual performance might draw the particular conclusion with respect to the prompting method into question. However, the prior proposed evaluation framework and algorithm are unaffected.

**Essential References Not Discussed:**

The paper is well embedded into current literature and comprehensively cites related works on causal reasoning abilities of LLM. Aspects on general reasoning of LLM and the role of compositionality within general deep learning models are discussed and cited accordingly. The probability of necessity and sufficiency and relations to potential outcomes seem to be sufficiently discussed and cited.

**Experimental Designs Or Analyses:**

The validity of Theorem 5.1 is empirically demonstrated via simulation in Appendix E. The experiment is suited to show and confirms the desired effects. The proposed "inductive CCR evaluation" algorithm follows naturally from the proposed definitions and metrics under the given multiplicative compositionality assumption.

The evaluation contains six LLMs of recent and moderate age, including of Llama 2/3.X, Phi-3-mini and GPT-4o. LLMs are prompted with and without CoT reasoning, showing expected differences in reasoning performance. CoT reasoning is realized via the provision of two example inferences. The provided CoT prompt wrapper (Fig. F.2) contains a strange number of quotation marks, which, however, shouldn't throw of newer models.

However, and to the best of my knowledge, realization of CoT via the provision of examples is rather uncommon for recent instruction-tuned 'chat' style models (e.g. GPT-4o or particular Llama-3.1-XX-Instruct variants) and better results are obtained with writing explicit instructions (which might again include example prompts). The 'incorrect' use of CoT prompting might explain the stark decreases in performance of GPT-4o in Fig. 7 for some scenarios (DY and XY). While authors explicitly do not aim to assess performance of the inspected LLM, they might still want to investigate the causes for the strong degradation of GPT-4o performance from 100% to 0% (or 2.5%, respectively), and confirm the correct prompting of instruction instruction-tuned 'chat' style LLMs.

The further discussion of evaluations and plots in Sec. 6.2 demonstrate the proper working of the proposed evaluation framework. Particularly, the increase in reasoning errors with an increasing number of mediator nodes aligns with findings of prior works.

**Methods And Evaluation Criteria:**

General approach on inspecting compositional causal reasoning abilities of reasoners is well motivated. The CCR aspects of validity and consistency (Def. 4.2 and 4.3) are soundly defined and natural candidates for the specific setup. The choice of ATE is common and the relation to PNS is explained and proven, making it a valid target metric. The proven PNS compositionality is shown over a simple yet sufficient graph.

Experiments on LLM where, again, tested over the synthetic graph (assumption due to Fig. 7 only featuring variables X,C,D and Y) and a constructed 'CandyParty' example, where variables in a chain depend on a given, fixed constant and previous variable value(s). While both setups are synthetic, they are sufficient to assess the proposed evaluation framework.

Although the paper only considers compositionality in a purely mathematical sense (e.g., requiring the full disentanglement of components and exclusion of compositional emergent properties), both setups are suited to show proper working of the proposed evaluation framework and the CCR task in general. The shown percentages of PNS estimates for individual components and the evaluation of validity, consistency and number of mediators in terms of (relative) absolute errors seems to be a natural choice.

**Other Comments Or Suggestions:**

1) Following Definition 5.1, the authors talk about the evaluation of "arbitrarily complex DAGs with cutpoints as if they were simply directed chains". While this statement is technically true in terms of the algorithm, the complexity now lies in computing the CCT edges. Due to the assumed compositionality, the authors might want to consider to mention the use of dynamic programming (or similar) techniques for computing the effects of individual CCT edges for more complex scenarios than the demonstrated ones.
2) I would assume consistency to generally imply a sort of systematicity within the model predictions, meaning that the predictions are 'consistently wrong'. Such behavior might usually defined as a (reasoning) bias. Depending on whether the authors agree on this view, they might consider mentioning this aspect in the paper.

**Other Strengths And Weaknesses:**

**Strengths**

1) The proposed metrics allow to assess the causal compositional reasoning abilities of LLM in a more fine-grained fashion than previous works that only consider the overall reasoning performance of LLM in terms of accuracy. The problem formulation and resulting metrics are well motivated and clearly defined.
2) The focus on the special case of multiplicative compositionality allows for a natural definition of the metrics and lets the proposed algorithm follows naturally. The validation of validity and consistency aspects is clearly separated and formalized under the given setting.
3) The experimental results are generally well presented and discussed. The experiments and simulations indicate the sound setup of the evaluation framework and the algorithm. Up to sparse outliers, the gained results indicate alignment with other findings of prior work.



**Weaknesses**

1) As already mentioned in the experimental designs section, I find the particularly strongly outliers within the results to be rather unusual. While the detailed evaluation LLM performance is mentioned as to not constitute the main focus of the paper, the proper working of the evaluation framework can only be shown within a valid setup. As mentioned before, the authors might improve on this topic by further investigating on the particular failure cases of the model and validate the correct application of CoT prompting for instruction-tuned chat-style LLM.
2) Within their taxonomy, the authors consider all four possible combinations of valid and consistent reasoners. While the authors argue that valid inconsistent reasoners should exist, I would consider the validity of ground truth to force answers to be consistent. Specifically, validity bounds every prediction $\hat\varphi_x$ to lie within $\pm\delta$ of the ground truth, such that valid but inconsistent predictions might, at most, lie $2\delta$ apart. While such samples might technically violate the imposed $\delta$-threshold, I would rather consider this a insufficiency of the proposed definition, than an inconsistency of the reasoner. This aspect might simply be improved by slightly extending the discussion in the paper.
3) The authors place their framework into the setting of multiplicative compositionality. Here, the forced reduction of CCT onto chains prevents the framework from considering the compositional (additive) effects of parallel components, (as, e.g., mentioned in the initial example of direct and indirect causal effects).

**Questions For Authors:**

My questions mainly regard the prompting and existence of valid inconsistent reasoners as mentioned before:

1) **Prompting.** Could the authors provide further insights on the particular cause of the reported outliers with regard to the proper application of CoT prompting?
2) **VI Reasoner.** Under which scenarios do valid, but inconsistent reasoners exist? Is the classification only due to technicalities of the definition or can they appear as a qualitative class of reasoners?

**Relation To Broader Scientific Literature:**

The authors soundly relate their approach to general (causal) LLM reasoning. The particular aspect of consistent compositional reasoning can be seen as a precondition for general causal reasoning. Most prior works in the field only considered the overall performance of causal LLM reasoning. The newly proposed metrics, therefore, allow for a more nuanced inspection on reasoning abilities in the often required joint reasoning over composed causal quantities. The measured LLM performances generally align with the findings of prior work.

Aspects on the probability of necessity and sufficiency and the relation of potential outcomes to ATE seem to be sufficiently discussed and proven.

**Theoretical Claims:**

I intuitively assume Theorem 5.1 in Appendix B.2 to be sound under the made assumption of monotonicity. However, I am not particularly familiar with the implications of applying potential outcome perspectives onto graphical models. The transformations within the presented formulas seem mostly to be consistent. Upon closer consideration it was unclear to me, how formulas B.20 -> B.21 simplify. E.g. why dropping the first and last terms of the addition is permitted (or rather why the respective terms have zero probability).

The introduced of commutative cut trees (CCT; Def. 5.2) is well defined and poses a practical aid to iterate the possible compositional combinations within the proposed algorithm.

While technically correct, the analysis of time complexity of the algorithm in Appendix D.1 might be expressed slightly more explicitly in the number of nodes only. From my understanding the considered setup reduces to paths between single source and sink nodes along a chain with all subpaths existing due to compositionality. Here, the problem might be formalized as selecting individual cut points from the superset of all cut points. By 'excluding' source and sink (but adding the fully compositional path), the number of compositional paths $k$ should be upper bounded by (or rater exactly be)  $2^{n-2} + 1$ for all $n>2$.

---

> ### Author Rebuttal · Authors · 2025-04-01
>
> We thank the reviewer for their thorough feedback and for acknowledging the soundness of our approach. We hope that the comments below address all concerns.
>
> 1. **“Outliers” for GPT-4o (CoT).**
>    - Results for $PNS_{XY}$ and $PNS_{DY}$ are not extreme outliers, though Fig 7 might imply such (we update to clarify this possible misinterpretation in-text). Please see original Figs F.7, F10: though the model did not achieve external validity according to our pre-defined threshold, predicted PNS distributions were not extremely far from truth.
>    - **Revision 1:** Added to Sec 6.2 (new figures https://tinyurl.com/bdddu2vx under `figures_and_tables/new_figures/error_analysis_figures/`): "Preliminary analyses indicate that GPT-4o with CoT displayed several kinds of faulty reasoning for both $PNS_{XY}$ (Figs. F.6, F.7) and $PNS_{DY}$ (Figs. F.8, F.9). These include (1) **failure to correctly extract causal relations** (e.g., true causal parents are missing from the stated parent set; Fig. F.9), (2) **incorrect logic despite correct causal relation extraction** (e.g., a clause will not logically follow from the preceding clause; Fig. F.8), and (3) a **truncated reasoning process**, where the model expresses its logic up to a certain point in the underlying causal graph but ignores the remainder of the graph without explanation (e.g., Fig. F.7)."
> 2. **CoT formulation.**
>    - The reviewer notes that “proper working of the evaluation framework can only be shown within a valid setup.” This is true, and **new results for o1** (which show complete VC reasoning) verifies that this setup is **valid and achievable by a sufficiently capable model.** See response to Xrrd.
>    - Potential influence of prompt engineering indeed motivated our comparison of CoT and non-CoT prompting. Our results for GPT-4o do indicate sensitivity to this. Though it is out-of-scope to perform additional prompt engineering experiments, this is an important area for future inquiry.
>    - **Revision**: Added under Limitations: "Future work should explore sensitivity to prompt engineering, including alternative formulations of CoT."
> 3. **Is VI reasoning a valid category?**
>    - **Revision:** As the VI reasoner is a counterintuitive yet informative category, we added the following to Sec 6.2: "This framework not only reveals *if* the reasoner is wrong, but offers insight into *how* it is wrong. We take VI reasoning as a stark and perhaps counterintuitive example, where external validity alone (as typically measured in reasoning evaluation) provides an incomplete picture. Consider GPT-4o with CoT: the model was near-valid to valid on the majority of quantities, implying some measure of success. Upon closer inspection, it was (near-)valid on most local estimates (e.g., $\widehat{\mathrm{PNS}}\_{CD}$) and their products (e.g., $\widehat{\mathrm{PNS}}\_{XC}\widehat{\mathrm{PNS}}\_{CY}$) but not for $\widehat{\mathrm{PNS}}\_{XY}$. While it might appear reasonable to assume that invalid global estimates stem from poor local reasoning or relation extraction abilities, the potential for alternative explanations is a key insight of our work. Here, GPT-4o displayed compositional inconsistency despite local relation extraction: it was more performant when asked directly about relatively proximal local relationships than when asked about distal cause-effect pairs where the answer relied on correct compositional reasoning over multiple direct and indirect relationships. VI reasoners highlight this potential failure mode: the reasoner does not learn to compose over multiple strands of logic yet can answer local questions when asked directly (akin to a student passing a math quiz simply by memorizing specific answers, instead of synthesizing; see incorrect logic despite relation extraction in new Fig F.8). Uncovering this behavior is precisely the benefit of measuring both external validity and internal consistency."
> 4. **Proof – B.20 $\to$ B.21.**
>    - We thank the reviewer for reading our proof line-by-line! This is due to monotonicity: $P(y′\_{x}) = 0$ and $P(z′\_{y}) = 0$ (outcome cannot be false when treatment is true). So, $P(y′\_{x}, y′\_{x′} )P(z\_{y′}, z′\_{y′})$ and $P(y\_{x},y\_{x′})P(z_{y},z′\_{y})$ zero out.
>    - **Revision:** Added to line in proof: “By monotonicity.”
> 5. **Potential outcomes vs graphical models.**
>    - This is not a problem, and going between the two is common (as they each provide their own usefulness). Note: “Formally, the two frameworks are logically equivalent; a theorem in one is a theorem in the other, and every assumption in one can be translated into an equivalent assumption in the other.” — Judea Pearl
> 6. **Complexity analysis.** We thank the reviewer for their attention and agree that this analysis can be improved.
>    - **Revision:** Notebook validating the improved complexity analysis: https://tinyurl.com/3tnut7tj
> 7. **Quotation marks in Fig F.2.** This was a copy-paste error. Updated figure: https://tinyurl.com/nhctxkat

---

> > ### Comment · Reviewer_doGN · 2025-04-02
> >
> > I thank the authors for comprehensively answering my questions. I mostly agree with the given perspectives and appreciate the additional results on the o1 model. Using a 'native' reasoning model eliminates possible problems of prompt engineering and CoT conditioning which where also mentioned by several other reviewers. The strong performance of o1 indicates that LLM might indeed be capable of performing causal reasoning within the specified setting. Considering that o1 now saturates scores within the conducted experiments, a natural and important next step would be the suggested extension to more complex graph setups. Given that earlier LLM weren't able to perform well in the experiments, might still present valuable insights for the current setup presented in the paper.
> >
> > Considering other reviews and rebuttals, I believe that many points have been well addressed. While pointing out the remaining point of low problem complexity, I believe that the presented work still gives interesting insights. I have therefore raised my score to a weak accept.

---

> > > ### Author Response · Authors · 2025-04-07
> > >
> > > We sincerely thank the reviewer for their continued engagement and for increasing their score to weak accept. Their comments have been very insightful. We highlight some follow-on comments below.
> > >
> > > ---
> > >
> > > ### **Implications of o1 saturation / failure of prior models**
> > >
> > > We agree with the reviewer overall. We will include a summary of the points made in their response in our revised text.
> > >
> > > As discussed, we have clarified in the revised text that our experiments were designed to **(1)** showcase the gamut of CCR error patterns (per our taxonomy of reasoners) and **(2)** validate that a correct implementation of Algorithm 1 can reveal these taxonomic distinctions. Thus, both **saturation and failure were desirable outcomes** in this case.
> > >
> > > We agree with the reviewer that the **observed failures do provide insights.** As the majority of models were invalid-inconsistent   (notably: Llama 3.1 Math, despite fine-tuning for math, and GPT-4o without CoT demonstrations, despite its size), this suggests that even simple CCR is a potential weakness in some LLMs, which warrants further investigation. Two additional points we make in the **revised discussion:**
> > > - **Small/open models:** We highlight that increasing CCR in relatively small LMs and open source models is an area worthy of attention in itself. In the revision: “Open source and smaller models did notably worse on this task than GPT-4o (CoT) and o1. This suggests  that even low-complexity CCR may be an area on which significant progress can still be made for an important class of LMs (smaller and/or open), an area worthy of attention in itself.”
> > > - **Impacts of CoT:** As noted in our response to Reviewer qJVr, error distributions on local and global quantities were significantly different for CoT vs non-CoT, except for o1 on $\mathrm{PNS}\_{XY}$ (Wilcoxon and t-test, alpha = 0.05; https://tinyurl.com/yck6bd28). And yet, CoT (in the current formulation) **did not consistently translate to improved validity and consistency**, as highlighted by our new Fig 9 (using CCTs as a visualization tool for error analysis; https://tinyurl.com/yc5cf9ev) and the reviewer’s useful points about **“outliers” in GPT-4o** with CoT.
> > >
> > > ---
> > >
> > > ### **Steps toward large-scale benchmarking**
> > >
> > > We update our limitations and future directions statement to acknowledge: “CCR experiments of scaling complexity (e.g., graphical and otherwise) and alternative prompt formulations (e.g., direct, formal causal queries as in Jin et al. 2023) will be an important next step for drawing robust conclusions on state-of-the-art causal reasoning in AI. This will also be essential for **probing the limits of CCR for o1** and more recent ‘reasoning’ models.”
> > >
> > > As summarized in our response to Reviewer Xrrd, the revised text will acknowledge our newly released **open source Python code for automated task generation**, which is designed to enable such benchmarking (see code and demonstration notebook:  https://tinyurl.com/5be5ehyp and https://tinyurl.com/4um56dab). More unit tests will be released at camera ready. Tasks take the general form presented in Section 6, but with random and user-controlled variability. To-date, this code provides the following.
> > >
> > > **Functionality provided to-date:**
> > > * Generate causal DAG.
> > > * Generate its corresponding CCT.
> > > * Randomly generate human names for nodes.
> > > * Enumerate quantities of interest for CCR evaluation: global, local, and compositions.
> > > * Generate text prompts corresponding to the generated SCM and chosen theme (factual, counterfactual).
> > > * Sample data from the SCM (observational, interventional).
> > > * Compute PNS, PN, PS, ATE from interventional data samples.
> > >
> > > **Prompt themes:**
> > > * CandyParty.
> > > * Football.
> > > * Flower Gardening (a **qualitative CCR task** that does not require numeracy).
> > > * Vaccination.
> > >
> > > **Sources of variation in the SCM:**
> > > * Total biconnected components in the causal DAG.
> > > * Nodes per biconnected component.
> > > * Graph type of biconnected component (cycle or wheel graph).
> > > * Parameter to Bernoulli distributions chosen uniformly at random or fixed.
> > > * Causal function (logical or, logical and; logical or is currently more reliable).

---

### Official Review · Reviewer_qJVr · 2025-03-13

**Overall Recommendation:** 3

**Summary:**

Regarding the combination of causal and combinatorial reasoning in generative AI, this paper presents a unified perspective called combinatorial causal reasoning (CCR), which is the ability to infer how causal measures are combined and how causal quantities are propagated in a graph. A framework for systematic evaluation of CCR is developed for average treatment effect (ATE) and probability of necessity and sufficiency (PNS). As a proof of concept, we demonstrate the design of CCR tasks on language models from the LLama, Phi, and GPT families, and experimentally show that CCR errors increase as the causal path complexity increases.

**Claims And Evidence:**

The main idea of this paper is to put forward a unified evaluation framework for combinatorial causal reasoning (CCR), and to prove the effectiveness of this framework through experiments.

Claim:
1. Combining causal reasoning and combinatorial reasoning through structural causal model (SCM) and combinatorial rules.
2. External validity (consistency with the true value) and internal consistency (consistency of the model itself) can measure the CCR capability.
3.  With the increase of causal path complexity, model error rate increases, and CoT hints are helpful for some models.

Evidence supports:
1. The authors demonstrate the composability of PNS in double-connected components (BCCs) through mathematical definitions and theorems (e.g. Theorem 5.1), providing a theoretical basis for the framework.
2.  Experiments on LLaMA, Phi, and GPT series models show a significant decline in the model's performance on complex causal pathways, and CoT suggests some help for GPT-4O.

**Essential References Not Discussed:**

na

**Experimental Designs Or Analyses:**

1. Using binary variables and logical "or" functions for SCMs is appropriate for monotonicity assumptions required by PNS/ATE compositionality (Theorem 5.1).    However, the choice of a single toy problem (CandyParty) limits generalizability to real-world causal scenarios with non-monotonic relationships or continuous variables.
2. Sampling 1000 exogenous variable sets per quantity is sufficient for distributional analysis, but the lack of statistical significance tests (e.g., t-tests for CoT improvements) weakens claims about CoT’s effectiveness.

**Methods And Evaluation Criteria:**

Methods:
1. The use of structural causal model (SCM) and directed acyclic graph (DAG) is a standard method in the field of causal reasoning, suitable for the decomposition and synthesis of combinatorial reasoning.
2.  The definition of composition as the functional relationship between global and local measures conforms to the tradition of probability graph model, but it may be too broad and lack the pertinence of specific tasks.
3.  Simplifying complex DAGs into chain structures through exchange tree cutting (CCT) is helpful for systematic evaluation, but some structural features of the original drawings may be ignored.

Evaluation criteria:
1. External validity and internal consistency can measure the accuracy and reasoning consistency of the model respectively, and are reasonable evaluation dimensions.
2.  Mathematical problems based on CandyParty can test counterfactual reasoning, but the problem complexity is low, and may not fully reflect the CCR ability in real scenarios.
3. The evaluation is only for ATE and PNS, and the causal function is assumed to be linear or monotonic, which may not be applicable to more complex causal relationships.

**Other Comments Or Suggestions:**

The commutative cut tree (CCT) transformation (Section 5.1) is a powerful abstraction but requires more detailed explanation to ensure readers grasp its role in simplifying complex DAGs.

**Other Strengths And Weaknesses:**

na

**Questions For Authors:**

1. The error analysis suggests that models struggle with longer causal paths due to "error propagation" and "losing the train of thought." Could you elaborate on the mechanisms underlying these failures? For example, are they due to limitations in attention mechanisms, parameter capacity, or task-specific prompt design?
2. The CoT prompting improved external validity for GPT-4o but not internal consistency. Do you have hypotheses about why CoT aids in local quantity estimation but fails to ensure compositional consistency? Could this be addressed through more sophisticated prompting strategies or architectural modifications?
3. The paper mentions releasing data and code upon publication. Could you provide a timeline or additional details about the reproducibility package (e.g., which components will be included, dependencies, etc.)?

**Relation To Broader Scientific Literature:**

Aiming at the problem that causal compositionality is not clearly and systematically combined in previous work, this paper proposes a formal expression of CCR, which defines combinatorial causal reasoning as the ability of model induction and deductive reasoning, and compensates for this problem through the establishment of a theoretical framework.

**Theoretical Claims:**

1. The proof in Appendix B.2 correctly shows that PNS composes multiplicatively across BCCs under monotonicity. The derivation leverages the monotonicity assumption to simplify joint probabilities, leading to the product formula. However, the proof assumes no unobserved confounders (Assumption A4), which may limit its applicability to real-world DAGs with latent variables.
2. The equivalence between PNS and ATE under monotonicity is derived correctly in Appendix B.3.1. By expressing ATE as a difference of probabilities and relating it to PNS via Equation (B.36), the authors establish a crucial link that justifies using ATE and PNS interchangeably for CCR evaluation.

---

> ### Author Rebuttal · Authors · 2025-03-31
>
> We thank the reviewer for their attention to the correctness of our proofs, the powerful abstraction provided by CCTs, and the gap in the literature that we attempt to close: the explicit and systematic evaluation of compositional consistency in causal reasoning. We address questions below.
>
> 1. **Low problem complexity.**
>     - As further emphasized by our **new results and discussion for o1** (see response to Reviewer Xrrd), we chose a relatively low complexity SCM to evaluate a range of models that we anticipated to reflect a spectrum of CCR behavior. The fact that our revised experiments demonstrate partial success (GPT-4o CoT), full success (o1), and failure (all other models) is important, as it validates that our framework is both correctly formulated and capable of disentangling distinct error patterns.
>    - We have **released automated task design code** that enables the user to generate CCR prompts based on casual graphs of scaling graphical complexity. An important future direction will be to benchmark on CCR tasks that simulate real-world problems (e.g., in healthcare), which the "vaccination" setting in our new task generation code is a first step toward.
> 2. **Parametric assumptions / estimands (ATE w. linearity and PNS w. monotonicity).** Though we instantiate our framework under select conditions in Sec 5, the framework presented in Sec 4 is highly flexible and is not limited to any estimand nor parametric assumptions. As stated in Sec 4.2, “Sec 5 suggests **one possible** procedure for assessing compositional consistency in causal reasoning (Alg. 1).” Under Limitations, we state “As this work only considers the ATE and PNS under Theorem 5.1, extensions could consider other estimands and compositional forms.”
>    - We provide Example 3.2 as one hint for future work: the decomposition of the TE under linearity. However, Pearl introduced a nonparametric mediation formula similar to Example 3.2 that could just as easily be exploited for CCR. Any nonparametric formula will be particularly suited to “realistic/real-world” task design. The possibilities are only limited by the extent of known compositional formulae in causal inference.
>    - **Revision:** Under Limitations: “Potential extensions could implement CCR tasks based on Example 3.2 or for Pearl’s nonparametric mediation formula, which extends Example 3.2 to nonlinear settings.”
> 3. **Unobserved confounding.** Sec 4 does not require that all confounders are observed, rather that valid causal inference can be performed. Sec 5 assumes this only to simplify task design.
>    - **Revision:** "Future work could accommodate latent confounding by exploiting known identifiability results for such cases." (See Pearl for many related works.)
> 4. **Inference hyperparameters.** Added to a new table in Appendix F.1.
> 5. **Elaboration on CCTs.** We thank the reviewer for acknowledging the utility of CCTs. We direct them to Figure D.1 of the original submission for more examples.
>    - **Revision 1:** Added to discussion w/ link to Fig D.1: "This allows for evaluation on arbitrarily complex DAGs with cutpoints as if they were simply directed chains (Fig. D.1)... As demonstrated in Sec 5.2, CCTs can be leveraged as a design tool for formulating reasoning tasks. As illustrated in Sec 6.2, they can also serve as an interpretable, intuitive visualization tool for graphically representing CCR successes and failures (Fig. 9)."
>    - **Revision 2:** New Fig 9 -- CCTs as a visualization tool for error analysis: https://tinyurl.com/yc5cf9ev
>    - **Revision 3:** We provide open source **code to generate CCTs from random DAGs** under sufficient conditions, along with our new automated task generation method: https://tinyurl.com/5be5ehyp and https://tinyurl.com/4um56dab.
> 6. **Mechanisms underlying failures / CoT and internal consistency.** Mechanistic explanations (e.g., limitations in attention mechanisms), additional prompting strategies, architectural modifications, or novel fine-tuning regimes (e.g., Huyuk et al 2025) are all potentially important areas for future work.
>    - **Revision:** While most of the above is out of scope for this work, we expand our error analysis for further insight. See our **response to Reviewer doGN.** Examples: https://tinyurl.com/bdddu2vx under `/figures_and_tables/new_figures/error_analysis_figures/`.
> 8. **Data and code release.**  All code provided in the repo above will be public, as will code used for **simulations** in Appendices C/E. Notably, this includes our **automated task generation codebase,** which enables customized prompt dataset generation. Code and data related to model inference will at minimum be available upon reasonable request after publication.
> 8. **Significance tests: CoT vs non-CoT.** See https://tinyurl.com/yck6bd28 for results. All error distributions for CoT were significantly different than non-CoT except for o1 on PNS(X,Y) (Wilcoxon, alpha = 0.05).

---

### Official Review · Reviewer_Xrrd · 2025-03-20

**Overall Recommendation:** 3

**Summary:**

The paper proposed a unified framework that evaluates compositional causal reasoning ability in large language models by measuring the average treatment effect and the probability of necessity and sufficiency.

**Claims And Evidence:**

The paper claims to introduce a framework for a comprehensive assessment of compositional consistency and demonstrates its utility through experimental results. However, the experimental results and setup explanation do not clearly illustrate the framework's usefulness or the specific insights it provides. Additionally, Figure 5 is quite confusing.

**Essential References Not Discussed:**

NA

**Experimental Designs Or Analyses:**

The experimental results presented in the main paper are limited. While additional results are provided in the supplementary material, the discussion and analysis are insufficient to convincingly demonstrate the usefulness of the proposed framework.

In particular, I found Figure 5 very confusing, and Figure 7 suggests that the task is not particularly challenging for GPT-4o with CoT. However, these results are presented without sufficient explanation or discussion of their implications.

Moreover, in the last paragraph of Section 6.2, the paper notes that errors increase as the complexity of paths increases—a rather obvious statement. If this is the main conclusion drawn from the experiment, it does not seem to add much value as a key result.

**Methods And Evaluation Criteria:**

The paper proposes evaluation metrics for assessing compositional causal reasoning, but it does not effectively explain the motivation behind such a framework. It is unclear why existing reasoning evaluation tasks are insufficient for evaluating causal reasoning ability.

Additionally, the explanation of the proposed method could be improved. The paper relies heavily on notations and definitions rather than concrete examples, making it difficult to follow. A rigorous mathematical definition of the metrics and quantities should be accompanied by examples to clarify their significance.

From my understanding, utilizing this framework requires access to the ground truth causal graph, which is often unavailable and difficult to obtain. Moreover, the paper presents only a framework without an accompanying dataset, which could significantly hinder its practical applicability.

**Other Comments Or Suggestions:**

NA

**Other Strengths And Weaknesses:**

The paper includes extensive background content in the supplementary materials, which helps readers understand the terminology and derivations.

**Questions For Authors:**

NA

**Relation To Broader Scientific Literature:**

The paper claims that the proposed framework explicitly and systematically incorporates causal compositionally. However, it does not clearly state the specific gap it addresses or aims to close.

**Theoretical Claims:**

Please see Methods And Evaluation Criteria.

---

> ### Author Rebuttal · Authors · 2025-04-01
>
> We thank the reviewer for their useful feedback. We hope that the following addresses your concerns.
>
> 1. **Limited results / no dataset.**
>    - **Revision 1:** CandyParty experiments now include **results for o1 with and without CoT.** For o1, CCR reasoning was “complete,” enriching our discussion by (1) validating the correct formulation of the task and (2) empirically demonstrating VC reasoning. Revised Figs 5/7 are at https://tinyurl.com/bdddu2vx under `/figures_and_tables/updated_figures/`.
>    - **Revision 2:** “Results for task T revealed three taxonomically distinct error patterns: VC (o1), near-VI (GPT-4o with CoT), and II (all remaining models). Only o1 placed mass in the VC reasoning quadrant (Fig. 5)... However, internal inconsistency made GPT-4o with CoT a near-VI reasoner: 75.1% of all estimates were near-valid,... and yet failure to correctly infer $\mathrm{PNS}_{XY}$ revealed internally inconsistent reasoning.” See our **response to Reviewer doGN** for further discussion and error analysis for GPT-4o. We hope these updates clear confusion on Fig 5, where each quadrant represents a reasoning type.
>    - **Revision 3:** To facilitate future benchmarking, we offer **open source Python code for automated task generation** for Algorithm 1 / Theorem 1. The user can scale the graphical complexity and choose from 4 themes: candy party, flu vaccination, football, and flower gardening. Further unit tests will be available at camera ready:  https://tinyurl.com/5be5ehyp and https://tinyurl.com/4um56dab
> 2. **Figure 7 / Task not challenging for GPT-4o with CoT.**
>    - As noted, this work is not meant as a benchmarking study but a validation of our framework. As further emphasized by our new results and discussion for o1, we chose a low complexity SCM to evaluate a range of models that we anticipated to **reflect a spectrum of CCR behavior.** The fact that our experiments demonstrate partial success (GPT-4o CoT), full success (o1), and failure (all other models) is important, as it validates that our framework is both correctly formulated and capable of disentangling distinct error patterns.
>    - **Revision:**  Added to concluding remarks: “Open source and smaller models did notably worse on this task than GPT-4o (CoT) and o1, suggesting that even low-complexity CCR is an area on which significant progress can still be made for an important class of LMs (smaller and/or open).”
> 4. **Motivation, usefulness, comparisons.**
>    - Novelty: (1) compositional consistency, (2) CCR evaluation in LMs, (3) metrics, (4) taxonomy of distinct error patterns, applicable beyond the causal setting, (5) practical algorithm.
>    - **Revision 1:** Added: “Comparison to Prior Methods: See Table A.1 for an extensive comparison. To further clarify our contributions, we compare to the CLADDER benchmark for formal causal reasoning (Jin et al., 2023). Most prior causal reasoning evaluation centers on commonsense knowledge and relation extraction (Table A.1). Like CLADDER, our framework and task generator (1) probe reasoning at the associational, interventional, and counterfactual levels; (2) apply causal inference methods; and (3) evaluate causal relation extraction. Our method notably departs from CLADDER in the following ways: (1) we evaluate compositional reasoning, another dimension of causal reasoning that has received little attention in LMs; (2) while CLADDER uses simple graphs (3-4 nodes), our task generator allows the user to scale graphical complexity; (3) we introduce new metrics for internal compositional consistency of causal reasoning; and (3) we do not require ground truth quantities, as the user can opt to evaluate on internal consistency only.”
>    - **Revision 2:** See Table A.1: https://tinyurl.com/4ctw6bfx.
> 5. **Required access to the true graph.**
>    - This is **standard in evaluation (see Corr2Cause, CLADDER, or arXiv:2402.11821)**. Evaluation innately requires reference values. The identifiability conditions for valid causal inference are extensively discussed in prior works. In general, these require **at least partial structural knowledge** (e.g., knowledge of confounders that block backdoor paths; of a valid instrumental variable when latent confounding is present; of mediators for the frontdoor adjustment; etc).
>    - This work is **explicitly rooted in causal graphical modeling** (rather than potential outcomes). We explicitly measure reasoning over graphical components (Theorem 1). Graphical reasoning is an important area itself. Thus, the **causal graph is centrally relevant.**
>    - **Revision:** Added to Sec 5.2: “When ground truth values are unavailable, Alg.1 can be run for internal consistency only.”
> 6. **“Metrics / quantities should be accompanied by examples.”**
>    - Each definition is accompanied by concrete examples related to Example 3.2, Fig 2, Fig 3 (e.g. "internally consistent reasoning entails that $\theta($..."). If the reviewer has specific confusions, we are happy to clarify further.

---

### Decision · Program_Chairs · 2025-05-01

**Decision:**

Accept (poster)

**Comment:**

The paper introduces a framework for evaluating compositional causal reasoning (CCR) in large language models (LLMs), focusing on metrics like average treatment effect (ATE) and probability of necessity and sufficiency (PNS). It claims to provide a systematic assessment of compositional consistency, validated through experiments on models such as LLaMA, Phi, and GPT-4o, showing increased errors with causal path complexity and partial benefits from chain-of-thought (CoT) prompting. Strengths include a clear theoretical foundation, rigorous proofs, and a taxonomy of reasoner types. Weaknesses noted by reviewers include limited experimental scope (small-scale, toy problems like CandyParty), unclear result presentation, reliance on ground truth causal graphs, and sensitivity to prompt engineering.

Following the authors’ rebuttal, reviewers raised concerns about motivation clarity, experimental generalizability, CoT prompt design, and the validity of the “valid-inconsistent” reasoner category. The authors addressed these by adding results for the o1 model, clarifying motivation through comparisons (e.g., to CLADDER), releasing open-source task generation code, and expanding error analyses (e.g., GPT-4o’s inconsistencies). In the AC-reviewer discussion, one of the reviewer increased scores after motivation and scope clarifications but sought better Figure 5 explanations. Another reviewer appreciating o1 results and framework validation, though noting low problem complexity. In my recommendation, I weighed the addressed novelty and theoretical rigor heavily, viewing the framework’s conceptual contributions as significant despite experimental limitations. The open-source code mitigates dataset concerns. The authors would need to improve the representation of some of the paper details and see if this can be applied to more complicated tasks.